# Bridging condensins mediate compaction of mitotic chromosomes

Giada Forte[1], Lora Boteva[2] , Filippo Conforto[1] , Nick Gilbert[2] , Peter R. Cook[3] , and Davide Marenduzzo[1]

**Eukaryotic chromosomes compact during mitosis into elongated cylinders—and not the spherical globules expected of self-attracting long flexible polymers. This process is mainly driven by condensin-like proteins. Here, we present Brownian-dynamic simulations involving two types of such proteins with different activities. One, which we refer to as looping condensins, anchors long-lived chromatin loops to create bottlebrush structures. The second, referred to as bridging condensins, forms multivalent bridges between distant parts of these loops. We show that binding of bridging condensins leads to the formation of shorter and stiffer mitotic-like cylinders without requiring any additional energy input. These cylinders have several features matching experimental observations. For instance, the axial condensin backbone breaks up into clusters as found by microscopy, and cylinder elasticity qualitatively matches that seen in chromosome pulling experiments. Additionally, simulating global condensin depletion or local faulty condensin loading gives phenotypes seen experimentally and points to a mechanistic basis for the structure of common fragile sites in mitotic chromosomes.**

## Introduction

During mitosis and meiosis, chromosomes condense into the iconic cylinders seen by light microscopy (Roberts et al., 2002; Marko and Siggia, 1997). Understanding how such cylinders form is a fundamental but unresolved question. Compaction into a cylinder instead of a sphere is surprising from the perspective of polymer physics, as polymers subjected to self-attraction usually collapse into spherical globules (De Gennes, 1979). Experiments show that fiber condensation is mediated by structural maintenance of chromosome (SMC) proteins, like condensins and cohesins, and disentanglement by topoisomerases (Swedlow and Hirano, 2003). While the contour length of loops remains unchanged as cells pass through mitosis (Jackson et al., 1990), rearranging them to form a long string of consecutive chromatin loops creates a bottlebrush polymer (BBP) with a large effective stiffness or persistence length, which is a prerequisite for cylindrical mitotic structures (Marko and Siggia, 1997; Goloborodko et al., 2016). Surprisingly, histone proteins, which are essential constituents of chromatin, are not required for cylinder formation (Shintomi et al., 2017). As both condensins and topoisomerases are ATP dependent (Swedlow and Hirano, 2003), it is normally assumed that active processes are required for condensation.

Mitotic condensation is thought to be largely driven by the action of two types of condensin: condensin II, which binds during prophase to form an axial scaffold, and condensin I, which is initially cytoplasmic and binds later during prometaphase to shorten chromosomes (Gibcus et al., 2018; Ono et al., 2003, 2004; Hirota et al., 2004). At the global level, the topologically associating domains (TADs), seen in interphase using a high-throughput variant of chromosome-conformation capture (3C) known as Hi-C, are typically lost during mitosis in minutes as the condensin II axial backbone may wind up into a helix (Gibcus et al., 2018). Notwithstanding this, imaging shows the gross structure of interphase chromosome territories is preserved in metaphase (Manders et al., 1999).

It has been suggested that the condensins that play such central roles in mitosis organize chromosomes locally in two distinct ways: by mediating the formation of stable and long-lived loops (Gibcus et al., 2018; Ono et al., 2003) and by bridging two different genomic segments to drive clustering (Hirota et al., 2004). While the latter bridging activity has not yet been fully established experimentally for condensins, it has been documented for other proteins in the SMC family, such as cohesins (Ono et al., 2004; Ryu et al., 2021). On the other hand, condensins are known to perform active loop extrusion in vitro (Ganji et al., 2018). The loop-extrusion model, which assumes that condensins are motors that can move on chromatin and extrude genomic loops, provides an appealing way to explain how the ATP-dependent activity of condensins can be harnessed to both generate and stabilize loops anchored to the backbone of

........................................................................................................................................................................................................

[1]School of Physics and Astronomy, University of Edinburgh, Edinburgh, UK;   [2]MRC Human Genetics Unit, Western General Hospital, Institute of Genetics and Cancer, University of Edinburgh, Edinburgh, UK;   [3]Sir William Dunn School of Pathology, University of Oxford, Oxford, UK.

Correspondence to Davide Marenduzzo: dmarendu@ph.ed.ac.uk;   Giada Forte: gforte@ed.ac.uk.

a mitotic chromosome (Goloborodko et al., 2016). Steric exclusion between different loops attached to the backbone then creates a large persistence length, which can be far greater than that of the underlying chromatin fiber and is a possible reason for the cylindrical appearance of chromosomes during mitosis and meiosis (Marko and Siggia, 1997; Paulson et al., 2021; Saitoh and Laemmli, 1994). While extremely useful as a starting point, this model still leaves many open questions. For example, the potential role of condensin-mediated bridging (rather than looping) is not directly addressed, and it remains unclear how further bottlebrush compaction might occur and whether it requires energy. Super-resolution and micromanipulation experiments suggest that condensins self-assemble into relatively inhomogeneous columns inside mitotic chromosomes (Sun et al., 2018), and the reason for this is unclear. Additionally, the elasticity of human mitotic chromosomes is striking as they can be stretched 10-fold by an external force and yet they relax back to their original length once the force is removed (Claussen et al., 1994); it is unclear to what extent this behavior can be recapitulated by existing models.

Here, we develop and characterize a simple polymer model to study chromosome compaction during mitosis. Significantly, our simulations do not involve constraining the polymer in a cylinder, as often done previously; then, the resulting shape emerges solely from specified interactions. We assume there are two types of condensin-like proteins with different activities. The first stabilizes loops (which provide an underlying bottlebrush geometry), while the second binds multivalently to chromatin to form local bridges. As mentioned above, the first role is well established (Ganji et al., 2018), but the second one remains speculative although proposed previously (Cheng et al., 2015; Kinoshita et al., 2022; Gerguri et al., 2021). We suggest such bridging may become particularly relevant after prophase when cytoplasmic condensin I associates with chromatin. We do not wish to suggest that condensin I solely acts as a bridge, but given its enhanced concentration at the onset of prometaphase, we argue that bridging by the newly added condensin I drives the striking morphological transition from the prophase bottlebrush to a shorter and stiffer mitotic cylinder. This compaction depends on the statistics and size of loops and topoisomerase activity. We also simulate the response of these structures to stretching, finding a qualitative behavior similar to that observed experimentally. Finally, we provide new insights into folding around common fragile sites (CFSs)—genomic regions of up to ~1.2 Mbp in which chromosomal lesions often appear following replication stress (Boteva et al., 2020).

Our model is simplified, as it includes only condensin and topoisomerase activities, and in part speculative, as it combines a looping and a bridging activity for both condensin I and II. Notwithstanding these limitations, it provides a parsimonious framework to rationalize qualitatively the existing experimental observations on mitotic chromosomes, in a way which, as we shall show, is impossible to achieve by assuming a single activity of condensin (either only looping or only bridging). Therefore, our model points to the crucial importance of condensin-mediated bridging in chromosome self-assembly; it also makes testable predictions, for instance, on the role of looping and bridging in the regulation of chromatin structure at fragile sites.

## Results
### A polymer model for mitotic chromosome folding
We studied the formation of mitotic chromosomes via coarse-grained molecular dynamics (MD) simulations (Fig. 1). A schematic of the model used is shown in Fig. 1. As experiments suggest mitotic chromosomes are arranged in consecutive loops (Naumova et al., 2013; Paulson and Laemmli, 1977; Marsden and Laemmli, 1979; Earnshaw and Laemmli, 1983) with contour lengths much the same as found during interphase (Jackson et al., 1990), we begin with a looped polymer depicted as a relaxed bottlebrush (Fig. 1, left). For simplicity, loops in this polymer do not change during simulations; we imagine their molecular anchors are provided by SMC proteins (we refer to these as looping condensins, Fig. 1, left) and that this loop configuration is created by active (Goloborodko et al., 2016) or diffusive (Brackley et al., 2017) loop extrusion. However, our focus is on the later folding dynamics driven by condensins, which can bridge (or bind multivalently) chromatin (we refer to these as bridging condensins, Fig. 1, right; Gibcus et al., 2018; Ono et al., 2003, 2004; Hirota et al., 2004; Sun et al., 2018; Marko, 2008).

We take into consideration cases where either all condensin-mediated loops have the same contour length $L_{loop}$ or the contour length varies following a Poisson distribution with an average value $L_{loop}$. Chromatin loops are composed of a sequence of beads with a diameter $\sigma$, which we assume to map to 20 nm, and to contain 2 kbp, in line with previous modeling work at this scale (Goloborodko et al., 2016; Rosa and Everaers, 2008). By using this mapping, the loop sizes we use are $L_{loop}$ = 80, 100, 120 *kbp* (see Materials and methods), in line with mitotic loops observed experimentally, which are usually 80–120 kbp long (Naumova et al., 2013; Jackson et al., 1990; Walther et al., 2018). Bridging condensins are modeled as diffusing beads (shown in green in Fig. 1) that bind reversibly and weakly to non-specific chromatin beads and strongly to the loop anchors (blue and red beads, respectively, in Fig. 1; see Materials and methods for force field used). This binding landscape constitutes an assumption of our model, which is inspired by chromatin immunoprecipitation experiments showing most proteins generically interact in different modes with DNA and chromatin (Roberts et al., 2002), such that there is a background of non-specific interaction (which we model by weak attraction) with peaks corresponding to specific interactions (which we model by strong attraction). It is reasonable that both looping and bridging condensins share specific interaction sites, which is why the strong interactions of bridging condensins are with loop anchors (where looping condensins are). We stress that no direct attractive interaction is specified between condensins (either looping or bridging). Typically, numbers of condensin bridges and loops are comparable, in line with previous modeling and experimental estimates (Goloborodko et al., 2016; Walther et al., 2018), and exact numbers do not qualitatively affect results.

The underlying chromatin fiber is characterized by a persistence length $l_p = 3\,\sigma \sim 60$ nm, which is consistent with that of interphase chromatin (Langowski, 2006). To model topoisomerase activity simply, chromatin strand passing is allowed as pairs of non-bonded polymer beads interact via a soft potential

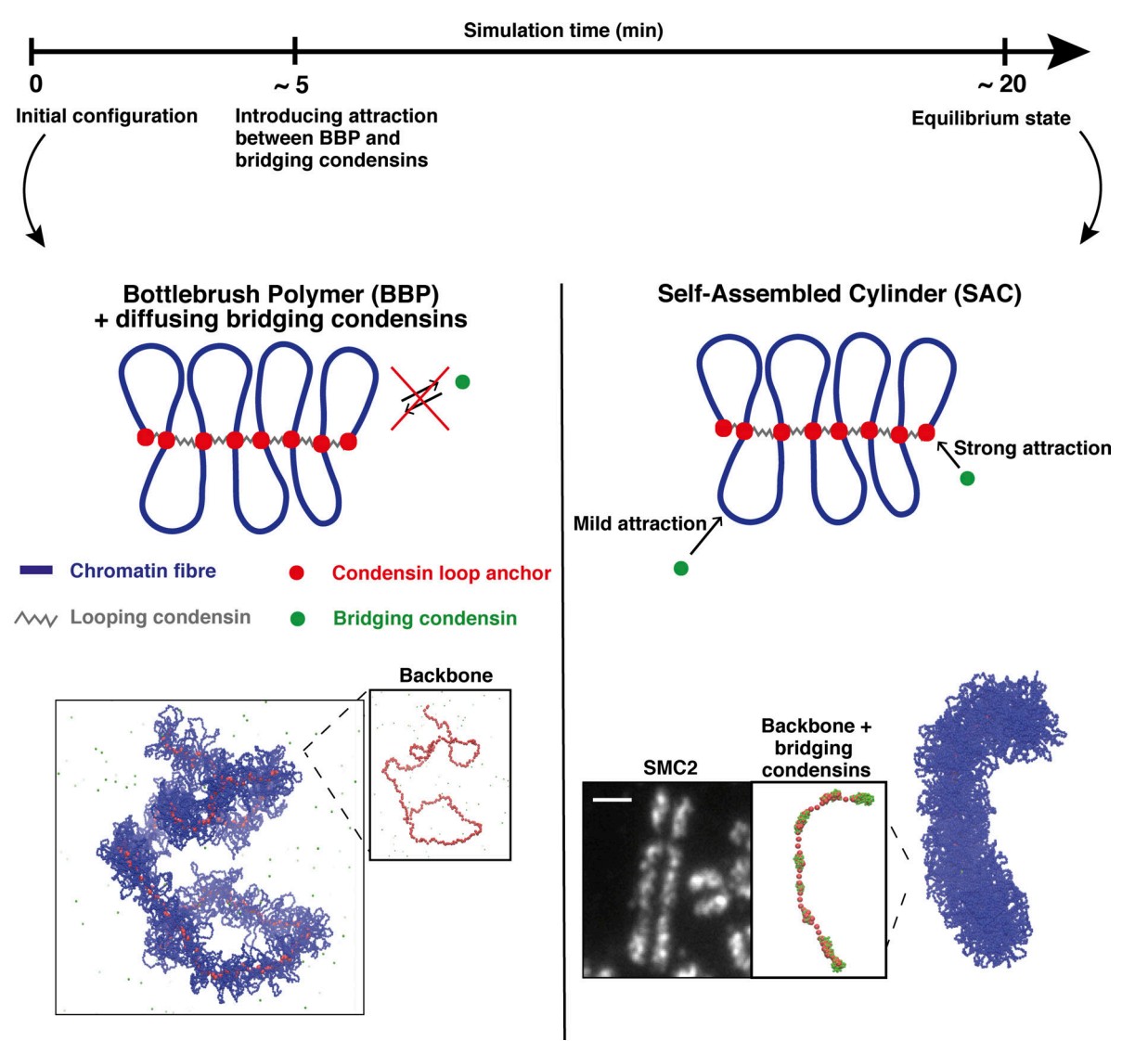

**Figure 1.** **Simulations of mitotic chromosomes.** Top: Simulation timeline. Center: Sketch of the model ingredients. Left: A string of blue beads (blue line) representing a mitotic chromosome is formed into a BBP with consecutive loops, which are assumed to be created by the action of looping condensins (gray segments, modeled as springs). Condensin loop anchors and bridging condensins are shown as red and green beads, respectively; the latter experience a purely steric interaction with the polymer in the first part of the simulation. Right: Starting from ~5 min, bridging condensins can bind reversibly to chromatin, weakly to blue beads, and strongly to red ones, generating a SAC. Bottom: Snapshots from computer simulations showing typical structures for the BBP (left) and SAC (right) regimes. The transition between the two is driven by condensin-mediated bridging. From left to right, the three insets correspond to backbone (condensin loop anchors) in the bottlebrush regime, an image of a mitotic chromosome from RPE1 untreated cells with SMC2 staining (scale bar = 2.5 μm, see Materials and methods for additional information), and backbone with condensin bridges in the SAC regime. The experimental SMC2 staining image reveals an inhomogeneous profile as emerges from our simulations.

$$U_{SOFT} = A \left[ 1 + \cos\left(\frac{\pi r}{r_c}\right) \right] \qquad (1)$$

where $r$ is the distance between two beads and $r_c = 2^{1/6}\sigma$ is a cut-off distance such that $U_{SOFT} = 0$ when $r > r_c$. The thermal energy of the system is $k_B T$, where $k_B$ is the Boltzmann constant and $T$ is the temperature. While the parameter $A$ does not have a straightforward mapping to experimental quantities, a change in its value efficiently represents a change in topoisomerase action. Therefore, when $A = 1\ k_B T$, polymer beads can cross each other because of thermal fluctuations, effectively modeling catalysis by topoisomerase II of strand cutting, movement of

another strand through the cut, and ligation. Increasing $A$ to $100\ k_B T$ models a reduction in topoisomerase activity, as in other works (Naumova et al., 2013; Brackley et al., 2015).

## Condensin-mediated bridging compacts bottlebrushes into cylinders

We first consider the case where all loops have the same contour length in an initial (pre-equilibrated) prophase-like state—the BBP configuration in Fig. 1, left. Starting from the BBP, attractive interactions are switched on between bridges and chromatin. A striking morphological transition now occurs (Video 1): the bottlebrush (Fig. 1, left) tightens up and becomes significantly

shorter and stiffer, forming a structure reminiscent of a metaphase chromosome (Fig. 1, right). We refer to this final configuration as a self-assembled cylinder (SAC). Compaction into a cylinder is driven by bridging, as bridges (like condensins) are assumed to be multivalent and able to bind more than one chromatin bead (or loop anchor; analysis of our trajectories shows that a condensin bridge typically has approximately three chromatin beads within a distance of 1.1 $\sigma$, or 22 nm). More specifically, the bridging-induced attraction (Brackley et al., 2013) provides a general mechanism to cluster bridges and to compact the polymer. It is based on positive feedback: bridging increases binding-site concentration locally, which recruits further bridges, and this triggers clustering. This would collapse a simple unlooped polymer into a spherical globule (Brackley et al., 2013), but here specific interactions with the loop anchors lead to the formation of microphase-separated clusters (Brackley et al., 2016). Additionally, in our case, competition between the bridging-induced compaction and looping-induced stiffening of the BPP drive self-assembly into cylinders.

Bridging condensins in SACs are non-uniformly clustered along the axial column—as seen in mitotic chromosomes in vivo (Sun et al., 2018; and Fig. 1, right inset). We suggest the nonuniform axial distribution is formed as a central condensin column breaks up in an effect akin to the Rayleigh instability: the bridging-induced attraction creates an effective surface tension, so when the interfacial energy becomes too large to maintain a contiguous column/stream, the column/stream breaks up into smaller globules. Loop size, $L_{loop}$, and soft repulsion, $A$, affect cluster size; the larger either is, the smaller the clusters are (Fig. S1).

We next quantify the geometric changes as the BBP morphs into a SAC in two ways (Fig. 2 A). First, the average gyration radius, $R_g$, was measured (Fig. 2, A and B, i and ii). $R_g$ sharply decreases once condensin binding begins (at $t$ = 0 in Fig. 2 B, i and ii); this is in accord with biological observations (Marko and Siggia, 1997). Interestingly, the extent of compaction depends on $L_{loop}$ and $A$; increasing $L_{loop}$ increases $R_g$ (Fig. 2 B i), while $R_g$ increases with decreasing topoisomerase II activity (Fig. 2 B ii). Consequently, strong topoisomerase activity (when $A$ becomes comparable to thermal energy) leads to more compaction. These results are consistent with the intuition that longer loops and stronger repulsion yield larger excluded volumes, preventing compaction, and exemplify another important role of topoisomerase II in chromosome folding.

Second, the acylindricity $Ac$ was analyzed by computing the length of the three eigenvalues of the gyration radius tensor, or equivalently the main axes $\lambda_1 \leq \lambda_2 \leq \lambda_3$ of the ellipsoid best approximating polymer shape (Fig. 2 A). If $\lambda_1 = \lambda_2 = \lambda_3$, the chromosome is spherical, while if $\lambda_1 = \lambda_2 < \lambda_3$, then it is an ideal cylinder (for mitotic cylinders the aspect ratio is >1). Acylindricity is defined as $Ac = \lambda_2 - \lambda_1$, and smaller values indicate a closer approximation to a cylinder. Condensin bridging reduces $Ac$, confirming that bridging renders the structures more cylindrical (Fig. 2 B, iii and iv). The role of topoisomerases is again apparent, as the smallest $Ac$ values are reached with the strongest topoisomerase activity (Fig. 2 B iv).

To quantify stiffness differently, we also computed the average tangent-tangent correlations between beads in different polymer segments. To do so, structures were coarse-grained and correlations were computed between vectors joining every fifth or tenth bead (to smooth effects of local crumpling of strings caused by condensin bridging, Fig. 2 B v). These correlations yield two main results (Fig. 2 B vi). First, binding stiffens structures (i.e., the correlation becomes larger), in line with the $Ac$ analysis and visual inspection of polymer snapshots. Second, in the starting BBP configuration, correlations are not monotonic and positive (as for worm-like chains; Marko and Siggia, 1995) at short distances, but often negative at intermediate distances (~1.2 μm along the backbone in the example shown in Fig. 2 B vi) to yield an oscillatory decay suggestive of a weakly helical nature for bottlebrushes. It is tempting to speculate that this effect is harnessed to create the narrow condensin II helices suggested by Hi-C data (Gibcus et al., 2018).

Cases studied thus far have loops with constant lengths; we now consider the more realistic situation where loops of average length $L_{loop}$ = 80, 100, 120 kbp were randomly generated according to a Poisson distribution with the desired average (Fig. 2 C). After switching on condensin binding, bridging again yields compact cylinders with nonuniform axial concentrations of condensins, albeit with a slightly more irregular cross-section due to the variability in loop size (Fig. S2). Gyration radius and acylindricity also change much as before (Fig. 2 C, i–iv). Again, the minimum $R_g$ and $Ac$ values were reached with the largest topoisomerase activity, while altering loop lengths had smaller effects. The radius of gyration and acylindricity were smaller in absolute value with respect to the uniform loop case. Clearly, Poisson-distributed loops therefore yield more compaction, and this can be understood in terms of the following simple calculation. Two bristles in a bottlebrush experience a repulsive force whose magnitude per unit of axial length can be estimated as (Marko and Siggia, 1997)

$$F_a \propto \frac{T}{\lambda}\left(\frac{Na}{\pi\lambda}\right)^{1/2}, \qquad (2)$$

where $T$ is the system temperature, $\lambda$ is the distance along the axis between successive bristles, $a$ is monomer size, and $N$ is the number of monomers per bristle. With uniform loops, $N = L_{loop}$, where $L_{loop}$ indicates the average loop size of a chromosome as above. However, with random loop size, the number of loop pairs with $N < L_{loop}$ is larger than the number of $N > L_{loop}$ because of the asymmetry of the Poisson distribution, and this brings down the total repulsive force for variable loops so that chromosomes become more compacted.

As simulations with variable loops avoid artefactual periodicities in contact patterns, we could use them to study how contact frequency varies as a function of genomic separation $s$. For intermediate distances, this frequency decays as $s^{-1/2}$ (Fig. 2 C vi). This is the same power law seen experimentally for 100 kbp < $s$ < 1–10 Mbp (e.g., see Fig. 2 C v, corresponding to late prometaphase, and Gibcus et al., 2018). While the power law seen in simulations holds for genomic distances smaller than ones observed experimentally, note that our polymers are shorter than real chromosomes and that additional compaction

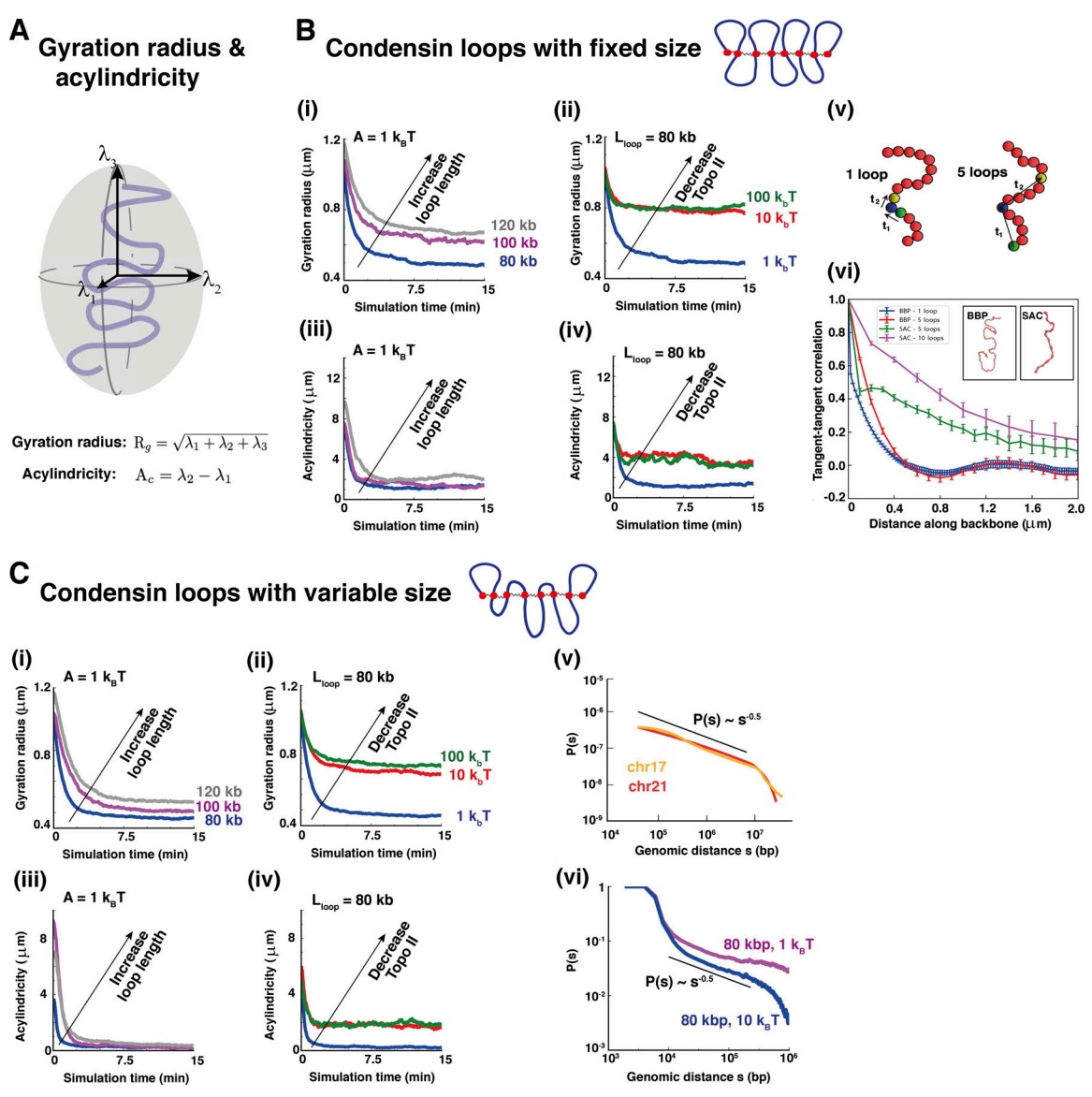

Figure 2. **Quantifying bridging-mediated chromosome compaction. (A)** Schematic showing the definition of gyration radius, $R_g$, and acylindricity, $Ac$. **(B)** Analysis of bridging-induced folding of mitotic chromosomes with fixed loop size. The time $t = 0$ corresponds to the instant at which condensin bridge binding is switched on. **(i and ii)** Temporal evolution of the gyration radius for simulations with fixed topoisomerase II activity strength $A$ and variable loop size $L_{loop}$ (panel i) and for simulations with fixed loop size and variable topoisomerase activity (panel ii). The values of $A$ and $L_{loop}$ are indicated in the figure. **(iii and iv)** Time evolution of the acylindricity for simulations with fixed $A$ and variable $L_{loop}$ (panel iii) and for simulations with fixed $L_{loop}$ and variable $A$ (panel iv). The decrease in loop length or increase in topoisomerase activity results in the acylindricity curves getting closer to zero, which indicates a more cylindrical shape. **(v)** Schematics of the coarse-graining procedure used to compute the tangent–tangent correlation of the chromatin fiber backbone. **(vi)** Resulting plots for a BBP and a SAC configuration ($L_{loop} = 80$ kbp, $A = 10$ $k_BT$). The x axis measures the distance along the backbone. Curves correspond to coarse-graining the backbones (as shown in the schematics in panel v) such that either 1 bead in 5 or 1 bead in 10 is considered ("5 loops" and "10 loops" curves). The coarse-graining does not much affect the results for BBP configurations, where the backbone is sufficiently smooth, but it has an effect for SAC configurations, as the backbone is locally crumpled in places—here coarse-graining is necessary to get a better estimate of the large-scale backbone bending. The negative dip for BBP structures is statistically significant (a two-sided Student test to see whether the minimum can be compatible with 0 returns a P value 0.002). The insets show snapshots of a BBP structure (left) and of a SAC structure (right). **(C)** Analysis of bridging-induced folding for mitotic chromosomes with variable size of condensin loops. Again, $t = 0$ corresponds to the instant when condesin bridge binding is switched on. **(i and ii)** Temporal evolution of the gyration radius for simulations with fixed topoisomerase II activity $A$ (panel i) and for simulations with fixed loop size $L_{loop}$ (panel ii). **(iii and iv)** Temporal evolution of the acylindricity for simulations with fixed $A$ (panel iii) and fixed $L_{loop}$ (panel iv). **(v and vi)** Experimental (panel v) and simulated (panel vi) contact probability between pairs of beads along the chromatin fiber versus genomic separation. Experimental curves refer to two different HeLa S3 mitotic chromosomes and are adapted from Naumova et al. (2013). The two simulation curves correspond to $(L_{loop}, A) = (80$ $kbp, 1$ $k_BT)$ (purple curve) and to $(L_{loop}, A) = (80$ $kbp, 10$ $k_BT)$ (blue curve). Once a couple of parameters $(L_{loop}, A)$ was fixed, all simulation curves in panels B and C were obtained by averaging over 10 independent simulations.

is likely to be achieved in late prometaphase in vivo with respect to our model (e.g., due to the onset of helicity suggested in Gibcus et al. [2018], which we do not capture here, and would presumably extend the validity of the power law). Importantly, simulations excluding the presence of chromatin loops cannot provide the power law mentioned above, indicative of the key role played by looping condensins and the bottlebrush structure (see Fig. S3).

While in this section single chromatids were considered, we also ran simulations of sister chromatids held together at centromeres (modeled by an additional set of springs joining the two sisters at the centromere). Then, condensin-mediated bridging plus topoisomerase action (modeled by a finite value of $A$ as before) leads to separation of the two sisters and compaction of each one (Video 2).

## Elasticity of SACs mirrors that of mitotic chromosomes

The mechanical properties of mitotic chromosomes have been investigated by micromanipulation experiments (Claussen et al., 1994; Houchmandzadeh et al., 1997; Poirier et al., 2000; Pope et al., 2006); for a simulation study complementary to ours, see also Ruben et al. (2023). Slow stretching can extend chromosomes by several times their length, yet they return to their normal size when allowed to retract. This indicates their internal structure is not significantly influenced by the applied force. (Above stretching forces of 20 nN, protein–DNA interactions break, leading to hysteresis in the extension–retraction cycle [Marko, 2008].)

Here, an extension–retraction cycle is simulated by applying constant and opposite pulling forces to the two ends of a SAC obtained at the end of the simulations discussed so far. A cycle has two steps: two equal and opposite forces, $\pm F$, are applied to the ends for ∼5 min when the cylinder reached its maximum extension (Fig. 3 A, left and center panels). Then forces are switched off and the polymer relaxed to reach a new equilibrium (Fig. 3 A, right). For concreteness, we focused on a single parameter set giving typical results: a fixed loop size $L_{loop}$ = 80 kbp and $A$ = 10 $k_BT$. Note that these simulations differ from micromanipulation experiments where one chromosome extremity is pulled at a constant and slow velocity, while the other remains fixed. Consequently, the two approaches are only equivalent in the thermodynamic limit (Titantah et al., 1999); nevertheless, we are mainly interested in the structural changes upon stretching and relaxation, and we expect these to be similar in the two cases. Pulling forces used in simulations vary in the 2 $pN$ ≤ $F$ ≤ 9 $pN$, and the large difference with those used experimentally is due to the fact that a single fiber was simulated, whereas in experiments many are effectively pulled simultaneously, as explained in Marko (2008). Note that the forces considered here are too small to dislodge histones from chromatin but strong enough to extend chromosomes.

During an extension–retraction cycle, cylinder extension is computed over time (Fig. 3 B). When the SAC is subject to the largest forces, the extension can increase fivefold and return to within 30% of its original value at the end of the cycle (Fig. 3 B and Video 3), much as is seen experimentally (Marko and

## A Extension-retraction cycle

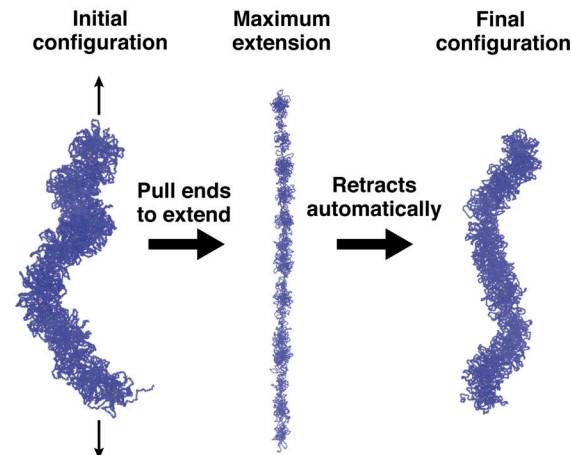

## B Chromosome extension versus time

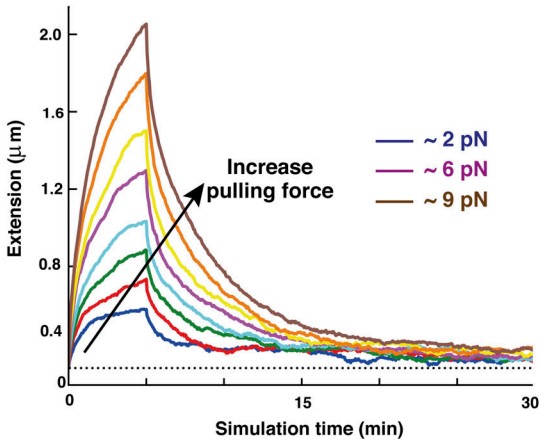

Figure 3. **Elasticity of SAC. (A)** Snapshots taken at the initial configuration (left), at maximum extension with $F$ ∼ 4 $pN$ (center), and after full retraction (right). **(B)** Changes in chromosome extension over time as the chromosome is first extended (for ∼5 min from its natural length [dotted horizontal line]) by different pulling forces $F$, before the pulling force is turned off and the chromosome relaxes. Loops have a fixed size $L_{loop}$ = 80 kbp and soft potential $A$ = 10 $k_BT$. Typical pulling force values are indicated in the legend. Each curve refers to a single simulation run.

Siggia, 1997). Noticeably, the action of bridging condensins appears to be crucial to obtain the latter result, as in simulations with only a looping activity chromosomes do not retract to their original extension (Fig. S4), possibly because the initial condition is a long-lived metastable structure that does not reform upon retraction. The moderate length difference seen between initial and final states in Fig. 3 points to a subtle difference in the structure before and after extension (Fig. 3 A, first and last configurations). Specifically, the initial configuration is a relaxed BBP with weak inherent helicity (Fig. 2 B vi) that is partially lost as the external force straightens the central axial column (see Materials and methods for more details).

## Simulating global condensin knockouts and local chromatin structure at common fragile sites

Having found qualitative agreement between simulation and experiment under normal conditions, we next evaluated the consequences of global and local perturbation of condensin activities. A global loss of looping condensins would yield the formation of multiple spherical condensin clusters and no cylindrical morphology, as the compaction due to bridging no longer competes with looping-induced stiffness in this scenario (see Fig. S3). Loss of bridging condensins instead would lead to a BBP structure (as in prophase). On the other hand, experimental knockouts of condensin I or II yield changes in the 3D interphase chromatin structure (Hoencamp et al., 2021; Brahmachari et al., 2022) as well as subtler mitotic phenotypes (Green et al., 2012), indicating that they are unlikely to have solely bridging and looping activities, respectively. Thus, condensin II knockouts have stretched chromosomes lacking axial rigidity; condensin I knockouts have wider and shorter fibers with a more diffuse backbone. To recapitulate these observations, we varied looping and bridging activities (Fig. 4). The condensin I knockout can be simulated with longer loops and fewer bridges (Fig. 4 B), consistent with the idea that any residual condensin II in the knockout yields longer loops. We predict that mitotic cylinders should become wider and shorter following such a global perturbation (Fig. 4 D). The condensin I knockout can instead be simulated by assuming that loops become shorter and bridging activity increases (Fig. 4 C). This is consistent with the idea that condensin I may work as a bridge or loop short chromatin regions; the resulting cylinders bend more locally and are thinner (Fig. 4 D), as found experimentally (Green et al., 2012). Simulations where the bridging activity is removed show how condensin I knockout structures can be recapitulated even by only considering changes in looping condensins. In contrast, including a change in bridging condensin activity appears to be fundamental to obtain the thinner chromosomes observed upon condensin II knockout (see Fig. 5).

While such global perturbations mimic experimental depletion experiments, the extent of condensin removal in the latter is difficult to quantify due to the importance of this protein complex for cell viability. Additionally, these perturbations are of limited relevance to mitotic chromosome structure in vivo. Instead, local perturbations, or defects, in condensin activity have been recently implicated as a mechanism to explain the appearance of CFSs, large genomic regions (up to ~1.2 Mbp in size) with increased likeliness of chromosomal lesions appearing after replication stress (Boteva et al., 2020). Our model and simulations can be used to test this hypothesis and predict what the consequences of faulty condensin activities might be on the local structure of metaphase-like SACs (Fig. 6).

First, complete loss of neighboring looping condensins was considered (simulating modeling faulty loading of both condensin I and II by removing two high-affinity-binding loop anchors; Fig. 6 A i). This led to a noticeable gap in the condensin backbone (Fig. 6 A ii), reminiscent of the lesions observed cytologically at some CFSs via DAPI staining (Fig. 6 A iii). Second, increasing the length of one condensin loop (simulating poor local condensin II recruitment; Fig. 6 B i) led to a different type

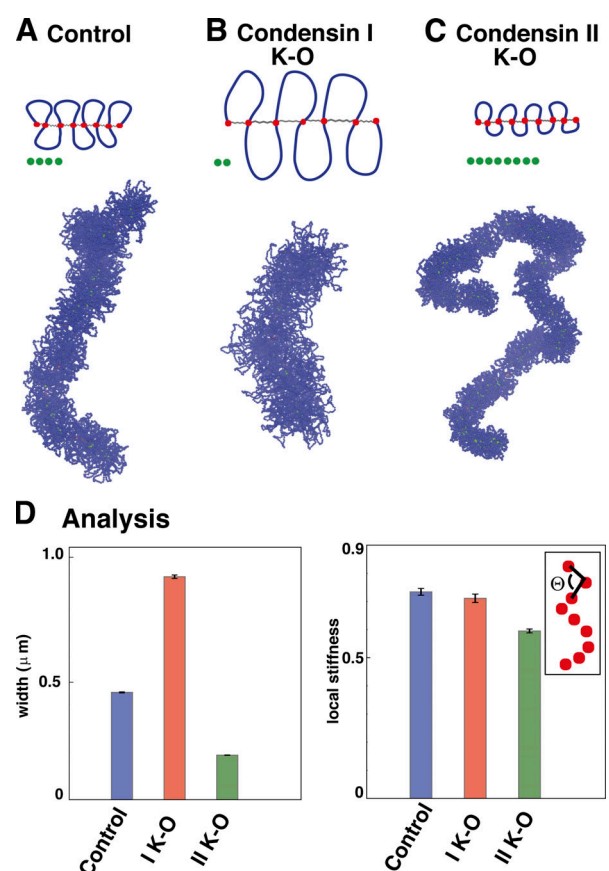

Figure 4. **Simulations of global condensin depletion. (A)** "Control" simulation with wild-type conditions, as in Fig. 1, SAC configuration. **(B)** Model setup (top) and simulation snapshot (bottom) for condensin I knockout (K-O)/depletion. We assume that the looping activity of condensin II (which remains after the knockout) leads to longer loops (gray arcs) and that the bridging activity (green beads) is smaller so there are fewer bridges. The fiber becomes wider and shorter. **(C)** Model setup (top) and snapshot (bottom) for simulations of condensin II knockout/depletion. We assume that the looping activity of condensin I (which remains after the knockout) leads to shorter loops, and that the bridging activity is larger so there are more bridges. **(D)** Quantitative analysis of width (left) and local stiffness (right) for the SACs in A–C. The local stiffness is computed by averaging the cosine of the $\theta$ angle between successive triplets of beads in the coarse-grained backbone (inset). The error bar is estimated by computing the mean over 10 independent simulation runs.

of defect, where the longer loop expands and is expelled out of the SAC, without creating any appreciable gap in the axial condensin backbone (Fig. 6 B ii). This resembles what is seen at other CFSs with fluorescence in situ hybridization (FISH) using two probes targeting adjacent chromosomal sequences—separation between fluorescent foci increases without appearance of any cytological lesion (Fig. 6 B iii). This concordance between the results of simulations and experiments is consistent with faulty condensin loading underlying the formation of CFSs. Additional simulations also highlight the key role played by bridging condensins without which it would not be possible to observe irregular structures such as cytological lesions (see Fig. 7).

It would be of interest to perform additional experiments to follow in more detail the path of the chromatin fiber in different types of molecular lesions to test our predictions more fully.

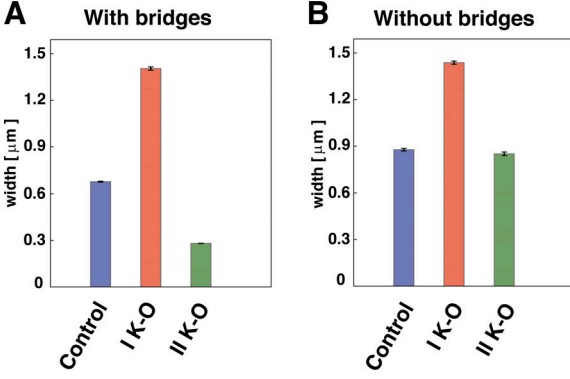

**Figure 5. Effects of bridging activity in condensin knockout simulations. (A and B)** Average mitotic chromosome width for control, condensin I, and condensin II knockout in simulations with and without bridging activity (A and B, respectively). In both scenarios (A and B), condensin I knockout was simulated with BBPs composed of longer chromatin loops, while condensin II knockout was simulated via shorter loops. The control and condensin I knockout cases do not seem very different between simulations with and without bridges. Instead, condensin II–depleted chromosomes do not change in width if condensin bridging activity is not included (see the "control" and "II K-O" bar in panel B), opposite to what is experimentally observed. For each of the six cases in panels A and B, the average and error bars are computed by employing 10 independent simulation runs.

## Discussion

In summary, we simulated the condensation of mitotic chromosomes in prometaphase (i.e., following prophase). The key assumption in our model is the simultaneous presence of two condensin activities: (1) a looping activity, leading to binding of two far chromatin sites to form stable loops (modeled via looping condensins viewed as springs), and (2) a bridging activity, modeling multivalent condensin–chromatin binding to form bridges between different regions of the fiber (modeled via bridging condensins viewed as diffusing spheres binding to chromatin through an attractive potential). The looping activity is well established experimentally, for instance through studies showing evidence of active loop extrusion activity by the condensin complex (Ganji et al., 2018). On the other hand, the bridging activity is more speculative and should be seen as a hypothesis, although based on the bridging activity found in other SMC proteins such as cohesins (Ono et al., 2004; Ryu et al., 2021); additionally, some previous models for condensin-mediated chromatin folding have included it (Kinoshita et al., 2022; Gerguri et al., 2021; Sakai et al., 2018). Another assumption we highlight is that bridging condensins bind weakly to all chromatin beads and strongly to loop anchors, which is where looping condensins are located. This assumption is based on the generic fact that chromatin-binding proteins (such as condensins) can bind either non-specifically (with a weak interaction) or specifically (with a strong interaction). The combined looping and bridging activity provides a key differentiator from previous polymer models (Goloborodko et al., 2016; Naumova et al., 2013; Gibcus et al., 2018) that traditionally just involve spring-like looping condensins that are either immobile (Gibcus et al., 2018) or continuously extrude loops (Goloborodko et al., 2016). In addition to considering shape changes, we also study

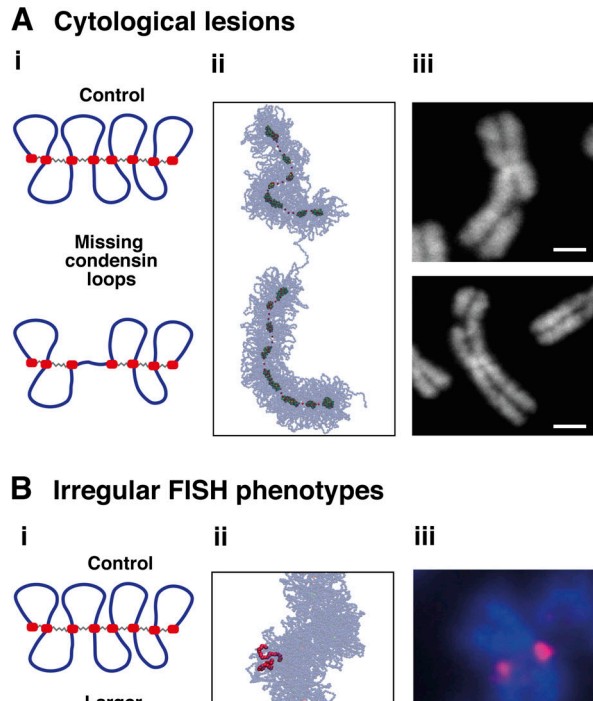

**Figure 6. Local condensin defects and mitotic chromatin structure at CFSs. (A)** Mechanistic model for cytological lesions. **(i)** Simulation setup investigating the consequences on chromatin structure of removing two condensin loops to model local depletion of condensins (the number of condensin bridges remains the same). **(ii)** Typical simulation snapshot. **(iii)** Images of mitotic chromosomes from RPE1 with DAPI staining after including replication stress with aphidocolin. Cytological lesions are visible (scale bar = 2.5 μm, see Materials and methods for additional information). **(B)** Mechanistic model for CFSs with irregular FISH phenotypes. **(i)** Simulation setup of another possible scenario associated with faulty condensin loading. In this case, three loops were joined together to create a single large loop (again condensin bridges remain the same): this scenario models faulty recruitment of condensin II, or in general of condensin looping activity. The violet segments mark the positions of the probes used in simulations to study how the FISH signal changes due to the perturbation shown. **(ii)** Typical simulation snapshot of the control case. **(iii)** Image of chromosomal defects visualized by FISH probes at a fragile site in control conditions in RPE1 cells (scale bar = 2.5 μm). **(iv)** Typical simulation snapshot for faulty condensin looping model. **(v)** Analogous FISH image at a CFS for RPE1 cells after depletion of the condensin component chromosome-associated protein H (CAP-H; scale bar = 2.5 μm). The simulated FISH probes appear to be close for the control case, while they separate when we model local faulty condensin looping, as in the experimental image.

chromosome elasticity plus condensin-associated defects at CFSs to provide orthogonal validation of our model.

Our main result (Fig. 1) is that condensin-mediated bridging can drive compaction of a prophase bottlebrush into a stiff self-assembled cylindrical structure, like that seen in prometaphase

## A Irregular FISH phenotype

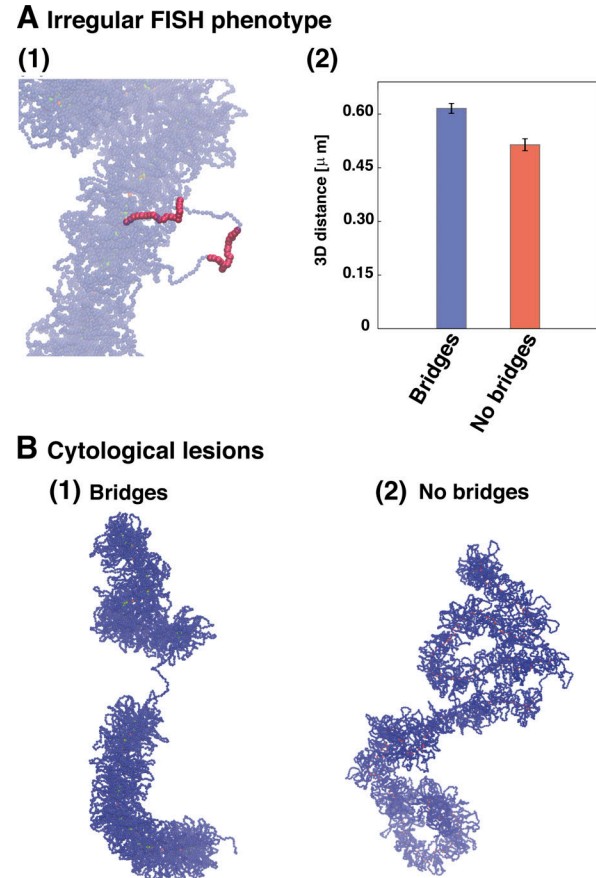

**Figure 7. Effects of bridging activity mitotic chromosome structure at CFSs. (A)** Analysis of bridging condensin effects on irregular FISH phenotypes, simulated by creating a longer loop within a regular bottlebrushed mitotic chromosome. 10 independent simulations with and without bridging condensins were performed, and the distance between two probes (violet segments in panel 1) was monitored. In the absence of bridges, the average 3D distance becomes slightly smaller (panel 2). **(B)** Study of the effects due to bridging mechanisms in the presence of cytological lesions. While the presence of bridges allows to clearly observe cytological lesions as in experiments (panel 1), their lack leads to a structure where the breakage is no longer visible (panel 2).

chromosomes, without energy input. While the bottlebrush geometry is well known from previous work (starting from Marko and Siggia [1997]) and provides a good starting structure of the prophase chromosome, we find that prometaphase compaction requires the additional presence of condensin-mediated bridging.

Besides their emerging shape, these SACs share several other features with real mitotic (prometaphase) chromosomes. First, in these simulations, bridging condensins are non-uniformly organized along the cylindrical axis (Fig. 1, bottom right), as seen in slightly stretched mitotic chromosomes (Sun et al., 2018); we suggest this is due to an effect like the Rayleigh instability that breaks up a contiguous axial column into smaller globules. Second, topoisomerase action (modeled via an effective soft potential to allow strand crossing) is important to form regular mitotic cylinders (Fig. 2), in line with long-standing experimental observations that topoisomerase plays a crucial

role in mitotic compaction in mitosis. Third, contact probability decays for intermediate genomic distance ($s$) as $s^{-0.5}$ (Fig. 2 C vi), in accord with Hi-C results (Naumova et al., 2013; Gibcus et al., 2018). Fourth, cylinder elasticity qualitatively mirrors that seen in the extension–retraction cycle of mitotic chromosomes (Fig. 3). Here, we predict that clusters of bridging condensins rearrange during stretching (Fig. S5), and this could be tested experimentally. Fifth, depleting looping and bridging activities in appropriate ways recapitulates both global phenotypes of condensin I/II depletion (Figs. 4 and 5) and local defects (Figs. 6 and 7) found at CFSs (i.e., defective looping gives large chromatid gaps and defective bridging leads to subtle increases in width; Boteva et al., 2020). These results are consistent with faulty condensin activity underlying CFS formation, with defective looping and bridging leading to different defects, which could be tested by inspection of stained condensin backbones at CFSs.

Importantly, we show that modeling only one type of condensin activity, whether looping or bridging, leads to results that are qualitatively inconsistent with at least some of the experimental evidence. Thus, including only the looping activity leads to incomplete compaction, incomplete retraction in simulated stretching experiments, and an inability to explain the formation of cytological lesions at common fragile sites. Similarly, including only the bridging activity leads to loss of cylindricity and to the inability to recapitulate phenotypes associated with condensin knockouts and structure seen at CFSs.

While the simplicity of our model renders the biophysical mechanisms underlying our observations more transparent, it also means that some potentially important ingredients have been disregarded. This is an inherent limitation of this type of work, and it points to ways for improvement. First, the axial condensin backbone in our cylinders lacks the helicity suggested by Hi-C results (Gibcus et al., 2018). While there is a weak helicity in the bottlebrush prior to compaction, additional ingredients are required to increase it during mitotic compaction. We note, though, that helices inferred from Hi-C are narrow, so an initial straight-line approximation may be acceptable. Second, our bridging and looping condensins are different species, whereas it is likely a single SMC protein performs both roles at different times. While one may expect that interchanges between looping and bridging modes should lead to qualitatively similar results, some key details may differ, and it would be useful to understand these (e.g., different condensin modes may become more relevant at different times in the cell cycle, for instance, as ATP activity decreases in late mitosis [Maeshima et al., 2018], and it would be desirable to quantify these in vivo). Third, the starting bottlebrush has consecutive loops (Fig. 1), but it would be of interest to study mixtures of nested loops that are more likely to be found in vivo (as there is some evidence that condensin I can create nested loops during metaphase [Gibcus et al., 2018]). Fourth, it would be instructive to study moving condensin springs (as in loop extrusion models [Goloborodko et al., 2016]) to see what effect movement adds to compaction driven by condensin-mediated bridging, and to further investigate a model where condensins can switch

between the looping and bridging activities. Fifth, to increase realism, it may be important to add the action of other mitotic proteins, for instance, those involved in the organization of the chromosome periphery (Booth and Earnshaw, 2017). Finally, while we concentrate on cylindrical mitotic structures, a different direction to explore would be to investigate whether models similar to ours can also help explain non-conventional liquid-crystalline chromatin arrangements found in interphase chromatin in specific organisms (Contessoto et al., 2023).

## Materials and methods

### Polymer physics modeling

In our simulations, a prophase chromosome is represented as a chain of beads organized like a BBP composed of consecutive loops. Each bead has size $\sigma$ in simulation units, corresponding to 20 nm or 2 kbp of the chromatin fiber (we use an intermediate value between a 10- and a 30-nm fiber and consequently a linear compaction slightly smaller than in works modeling a 30-nm fiber [Brackley et al., 2017]). A sketch of the model is shown in Fig. 1 of the main text.

Interactions between polymer beads are described by four potentials. First, non-adjacent beads interact via a soft potential defined as

$$V_{soft}(r_{i,j}) = A\left[1 + \cos\left(\frac{\pi r_{i,j}}{r_c}\right)\right],$$

where $r_{i,j} = \left|\vec{r_i} - \vec{r_j}\right|$ is the distance between the i-th and the j-th bead, $r_c = 2^{\frac{1}{6}}\sigma$ is a cutoff distance, and $A$ is a parameter defining the strength of the repulsion between two beads that we set equal to 1, 10, or 100 $k_B T$, where $k_B$ is the Boltzmann constant and $T$ is the temperature of the system.

Second, adjacent beads are connected via a harmonic potential $V_{harm}$ whose expression is the following

$$V_{harm}(r_{i,j}) = K(r_{i,j} - r_0)^2,$$

with $r_0 = 1.1\ \sigma$ being the equilibrium distance and $K = 100\ \frac{k_B T}{\sigma^2}$ the spring stiffness.

Third, the polymer is characterized by a bending rigidity described by the Kratky-Porod potential

$$V_{BEND}(\phi_i) = K_{BEND}(1 + \cos\phi_i),$$

where $\phi_i$ is the angle between beads $i$–1, $I$, and $i$+1, while $K_{BEND} = 3\ k_B T$ defines the filament rigidity and corresponds to a persistence length $l_p \sim 60$ nm compatible with the persistence length of chromatin (Langowski, 2006).

Finally, additional springs are inserted to create the bottle-brush loops by connecting a loop anchor $i$ (red beads in Fig. 1) with the chromatin bead immediately preceding the next loop anchor along the polymer. The distance $|j\text{-}i|$ corresponds to the loop length. The potential describing these springs is the following:

$$V_{backbone}(r_{i,j}) = K_{backbone}(r_{i,j} - r_1)^2,$$

where $K_{backbone} = 100\ \frac{k_B T}{\sigma^2}$ and $r_1 = 1.8\ \sigma$.

To study the compaction of mitotic chromosomes via condensin-like bridges, we insert additional spheres diffusing in the simulation box and experiencing an attractive interaction with the polymer, which is modeled by a truncated Lennard-Jones potential

$$V_{LJ,cut}(r_{i,j}) = 4\varepsilon\left[\left(\frac{\sigma}{r_{i,j}}\right)^{12} - \left(\frac{\sigma}{r_{i,j}}\right)^6\right]\Theta(r_c - r_{i,j}),$$

where $r_c = 2^{\frac{1}{6}}$ is the cutoff distance while $\varepsilon$ defines the strength of the attraction. We set $\varepsilon = 3\ k_B T$ between bridges and generic polymer beads and $\varepsilon = 8\ k_B T$ between bridges and loop anchors. Additionally, bridges interact with each other via steric interactions described by the potential $V_{LJ,\ cut}$, with $\varepsilon = 1\ k_B T$.

All simulations with a single sister chromatide contain between 300 and 400 chromatin loops with average size $L_{loop}$. The latter together with the topoisomerase II activity strength $A$ determines the parameters of our model. Each simulation is then characterized by a couple, $(L_{loop}, A)$, whose values can be $L_{loop} = 40, 50, 60\ \sigma = 80, 100, 120$ kbp, and $A = 1, 10, 100\ k_B T$.

In the initial setup, the number of condensin bridges is $N = 500$ comparable to the number of loops. Instead, when we simulate condensin I or II depletion, we halve or duplicate $N_p$, respectively. Finally, for simulations involving the self-assembling of sister chromatids, we use $N_p = 1,000$.

### Langevin dynamics

The dynamics of polymer and protein bridges are described by the Langevin equation

$$m_i\frac{d^{2r_i}}{dt^2} = -\nabla_i U - \gamma_i\frac{dr_i}{dt} + \sqrt{2k_B T\gamma_i}\eta_i,$$

where $m_i$ is the mass of the $i$-$th$ bead, $r_i$ its position, $U$ is the total potential energy of the system, and $\gamma_i$ is the friction coefficient. Finally, $\eta_i$ is the stochastic Brownian noise whose components respect the following equations

$$<\eta_i(t)> = 0\ \text{and} <\eta_{i,\alpha}(t)\eta_{j,\beta}(t')> = \delta_{i,j}\delta_{\alpha,\beta}\delta(t - t'),$$

where $\delta_{i,j}$ is the Kronecker delta and $\delta(t - t')$ is the Dirac delta function.

The Brownian dynamics is simulated through the LAMMPS software (Plimpton, 1995) by using a time step $dt = 0.01\ \tau_{LJ}$ with $\tau_{LJ} = \sigma\sqrt{\frac{m}{k_B T}}$. For a polymer bead, we set its diameter $\sigma$, energy $k_B T$, and mass $m$ equal to 1 in simulations units. There are two other timescales in the system besides $\tau_{LJ}$, namely the velocity decorrelation time $\tau_{dec} = \frac{m}{\gamma}$ and the Brownian time $\tau_B = \frac{\sigma^2}{D_B}$. By setting the friction $\gamma = 1$, we get $\tau_{LJ} = \tau_{dec} = \tau_B = 1$ as $D_B = \frac{k_B T}{\gamma}$. To map times from simulation units to real units, we use $\tau_B$. From the Stroke–Einstein equation for spherical beads of diameter $\sigma$, we know that $\gamma = 3\ \pi\sigma\eta_{sol}$, where $\eta_{sol}$ is the solution viscosity. We then get $\tau_B = \frac{3\pi\sigma^3\eta_{sol}}{k_B T}$. By setting $\sigma = 20\ nm$ (or equally 2 kbp), $T = 300\ K$, and $\eta_{sol} = 150\ cP$, which is reasonable for the nucleoplasm, we finally found $\tau_{LJ} = \tau_B \sim 3\ ms$. Finally, the mapping of the pulling forces used in extension–retraction simulations can easily be obtained through the quantities already introduced. In LJ units, the force is given by $F = \frac{k_B T}{\sigma}$. By using $T$ and $\sigma$ mentioned

above, we obtain that the simulation force unit corresponds to $F = \frac{k_B T}{\sigma} \sim 0.2 pN$.

## Analysis of clusters of condensin bridges

Here, we provide the results of a cluster analysis performed on condensin bridges.

First, we investigate how clusters of bridging condensins change during mitotic chromosome folding depending on the loop size $L_{loop}$ and the strength of the soft potential, i.e., of the topoisomerase action, $A$. In Fig. S1, we show the results of the cluster analysis while mitotic chromosomes with fixed loop size fold due to the attractive interaction with bridging condensins. We note that a small soft potential (i.e., a weaker excluded-volume repulsion) leads to the formation of fewer and bigger protein clusters.

Second, we perform a similar analysis for simulations reproducing micromanipulation experiments where mitotic chromosomes are pulled and released to investigate their elasticity. In this kind of simulation, we apply a pulling force to the two extremities of the chromosome backbone (formed by the red beads in Fig. 1) and, after the chromosome has been stretched up to five times its original length, we remove the pulling force and let the chromosome relax to the equilibrium condition. In Fig. 3 B, we observe that, at the end of the extension–retraction cycle, the cylinder length is slightly larger than the initial one and we can wonder if this is an effect due to a change in clusters of bridging condensins. In Fig. S5, we plot the average cluster size and the number of clusters during a whole extension–retraction cycle at different pulling forces. We see that small forces (i.e., $F < 6$ $pN$) are too weak to disrupt clusters (see Fig. S5, top left), presumably because they do not stretch the cylinder enough (see Fig. 3 B). Instead, for larger forces (see Fig. S5, top right panel and two bottom panels), clusters reduce in size and increase in number during the extension step ($0 \leq t \leq 5$ min) and merge again when the pulling force is switched off ($t \geq 5$ min). Therefore, even if small forces do not have any effects on clusters of condensin bridges, they are strong enough to slightly stretch the cylinder reducing its original (weak) helicity, which is not re-established during the retraction step. This means that the conformation prior to stretching (with weak helicity inherited from the bottlebrush structure) is a long-lived metastable configuration.

## Experimental methods
### Cell preparation
RPE1 (female) cells are cultured in Dulbecco's Modified Eagle Medium F12 (Cat No. 12500-062; GIBCO) supplemented with 10% fetal calf serum, 1% Pen-Strep, and 1% L-glutamine. Growth media for RPE cells contains 0.3% (wt/vol) sodium bicarbonate (Cat. No. S5761; Sigma-Aldrich). Cells are maintained at a temperature of 37°C in an atmosphere of 5% $CO_2$. RPE1 cells are subjected to regular mycoplasma testing and cell authentication is performed via karyotyping.

### Replication stress induction
Cells are synchronized at the boundary between G1 and S phases by adding high-dose aphidicolin (APH; Calbiochem). Media containing 5 mg/ml APH is added to cells for 2 h to block cell cycle and stop cells at the G1/S boundary. Cells are washed in PBS and released in normal growth media. Cells are observed to progress synchronously from S phase into G2 through FACS analysis and immunofluorescence of cell population at 2–10 h following release. Replication stress is induced by low-dose treatment of APH (0.4 mM APH) for extended periods (12–24 h).

### Preparation of human metaphase chromosomes
RPE1 cells are treated with 0.1 mg/ml colcemid (Cat. No. 15210-040; Life Technologies) for 1 h prior to harvest to induce mitotic arrest and increase the number of mitotic cells. Cells are trypsinized and washed in PBS. Hypotonic solution, which contains 75 mM KCl, is added dropwise to reach a final 5 ml volume. Hypotonic treatment is performed at RT for 10 min, after which cells are pelleted by centrifugation at 1,200 rpm for 5 min and fixed three times in 5 ml of freshly prepared solution of 3:1 ratio (vol/vol) methanol:acetic acid (MAA). The MAA fixative is added to the cell pellet dropwise with constant agitation and the prepared chromosomes are stored at 20°C. Metaphase chromosome preparations are dropped onto a glass slide to prepare slides with metaphase spreads. The glass slides are pretreated in a dilute solution of HCl in ethanol for at least an hour prior to use. The chromosome preparations are pelleted by centrifugation at 1,500 rpm for 5 min and resuspended in freshly prepared MAA solution until the suspension becomes cloudy. Two drops of the suspension are dropped onto a pretreated glass slide from a height of 20 cm and dried at RT overnight before staining or hybridization.

### Cytogenetic analysis of common fragile sites
Two approaches are used to map the location of common fragile sites. First, a visual inference of the fragile locus position is inferred by using reverse DAPI banding. Second, the position of the fragile site is determined by computing the distance along the chromosome arm. Through this approach, the total length (a), in pixels, of the chromosome arm where the break occurs, and the pixel length of the distance between the centromere and the break (b) are measured. The ratio (b)/(a) is calculated and used on scaled models of banded chromosomes (from the International System for Human Cytogenetic Nomenclature) to infer the breaks' genomic locations. The ratios focus along the chromosome arms, suggesting recurrent breaks at CFS locations, and the midpoint of each cluster is taken as a putative CFS location. As fixation and spreading of chromosomes might cause some distortion, FISH methods are employed to perform a molecular fine-mapping of the most frequent CFS regions.

### FISH
DNA is prepared from the BACs or Fosmids and labeled. Probes are labeled using a nick translation reaction with the uridine analogs biotin-16-dUTP (Cat. No. 11093070910; Roche) or digoxigenin-11-dUTP (Cat. No. 11093088910; Roche). Nick translation is performed in a 20-ml reaction volume containing 1–1.5 mg DNA with 5 ml each of 0.5 mM dATP, dCTP, and dGTP and either 2.5 ml of 1 mM biotin-16-dUTP or 1 ml of 1 mM digoxigenin-11-dUTP. DNase I (Cat. No. 4716728001; Roche) is

added to a final concentration of 1 U/ml, and DNA polymerase I (Cat. No. 18010025; Invitrogen) is added to a final concentration of 0.5 U/ml. The reaction is performed in 1× nick translation salts (NTS) buffer containing 50 mM Tris, pH 7.5, 10 mM MgSO$_4$, 0.1 mM DTT, and 50 mg/ml BSA for 90 min at 16°C. Unincorporated nucleotides are removed by gel filtration of the NTS reaction through a G50 Sephadex spin column (Cat. No. G50DNA-RO; Roche). Slides, which contain either MAA-fixed chromosome spreads or PFA-fixed nuclei, are treated with 100 mg/ml RNaseA (Cat. No. 12091039; Invitrogen) in 2× SSC for 1 h at 37°C, washed briefly in 2× SSC, and dehydrated through an ethanol series (2 min each in 70%, 90%, and 100% ethanol). Slides are air-dried and baked at 70°C for 5 min before denaturation, which is performed in 70% formamide (vol/vol) in 2× SSC (pH 7.5). Slides containing MAA-fixed chromosome spreads are denatured at 70°C for 1 min, while slides on which cells were cultured and then fixed in 4% PFA are denatured at 80°C for 20 min. Following denaturation, slides are submerged in ice-cold 70% ethanol for 2 min and then dehydrated through 90% and 100% ethanol for 2 min each at RT. For hybridization, 150 ng of the labeled probe is combined with 5 mg of salmon sperm and 10 mg of human Cot1 DNA (Cat. No. 15279011; Invitrogen). Two volumes of ethanol are added and the probe mix is collected by centrifugation and dried. Dried probes are resuspended in 10 ml of hybridization buffer containing 50% formamide (vol/vol), 1% Tween-20, and 10% dextran sulfate (Cat. No. D8906-100G; Sigma-Aldrich) in 2 × SSC. Probes are denatured at 70°C for 5 min and reannealed at 37°C for 15 min and chilled on ice. Probes are pipetted onto slides and hybridization was performed at 37°C overnight. Coverslips are then removed and slides are washed four times in 2× SSC at 45°C for 3 min and four times in 0.1× SSC at 60°C for 3 min. Slides are then blocked in 5% milk in 4× SSC for 5 min at RT. Biotin labels are detected with sequential layers of fluorescein (FITC)-conjugated avidin (Cat. No. A-2011; RRID: AB_2336456; Vector Labs), biotinylated anti-avidin (Cat. No. BA-0300; RRID:AB_2336108; Vector Labs), and a further layer of FITC-avidin. Digoxigenin is detected with sequential layers of Rhodamine-conjugated anti-digoxigenin (Cat. No. 11207750910; RRID:AB_514501; Roche) and Texas-Red (TR)-conjugated anti-sheep IgG (Cat. No. TI-6000; RRID:AB_2336219; Vector Labs). Slides are DAPI stained, mounted in Vectashield, and imaged on a Zeiss epifluorescence microscope with a 100× objective. Data are collected using micromanager software and the analysis is performed through scripts in iVision or ImageJ.

### Immunofluorescence

For immunofluorescence on metaphase chromosomes, we fixed cell suspensions in a 3:1 methanol:acetic acid solution, dropping them onto glass slides. After allowing incomplete drying, we promptly immersed them in PBS for 5 min at RT. Following this, slides underwent a wash in TEEN buffer (10 mM Triethanolamine-HCl, pH 8.5, 2 mM EDTA, 250 mM NaCl) and were blocked in 10% fetal calf serum at 37°C for 10 min. Primary antibodies (from Kumiko Samejima, Wellcome Centre for Cell Biology, The University of Edinburgh, Edinburgh, UK; Boteva et al., 2020) were applied at the required dilutions and left to incubate in a humidified chamber at 37°C for 30 min. Subsequently, slides were washed in KB buffer (100 mM Tris-HCl, pH 7.7, 1.5 M NaCl, 1% BSA). For the next step, secondary antibodies, derived from donkeys and conjugated to fluorophores (Jackson Immuno Research), were diluted to 1:500 in TEEN buffer, added to the slides, and incubated at 37°C for 30 min. After another wash in KB buffer, staining with 50 µg/ml DAPI for 3 min at RT was conducted to detect DNA and nuclei. Finally, slides were mounted in Vectashield (Cat. No. H-1000; Vector Laboratories) and subjected to imaging on a Zeiss Epifluorescence microscope using a 100× objective. The anti-SMC2 antibody was detected using a FITC-labeled anti-rabbit secondary antibody at a 1:500 dilution (Cat. No. 711-095-152; RRID:AB_2315776; Jackson Immuno Research).

### Online supplemental material

Fig. S1 shows clustering analysis for bridging condensins during mitotic chromosome compaction; Fig. S2 shows snapshots of mitotic chromosomes with Poisson-distributed loops; Fig. S3 shows analysis of the contact probability in the absence of chromatin condensin loops; Fig. S4 shows mitotic chromosome extension during an extension–retraction cycle; Fig. S5 shows cluster analysis for condensin bridges during an extension–retraction cycle. Video 1 shows the self-assembling of a mitotic chromosome mediated by bridging condensins; Video 2 shows bridging-mediated compaction of bottlebrush sister chromatids; Video 3 shows that bridging-mediated folded chromosomes are elastic objects.

### Data availability

The code used for the simulation is LAMMPS, which is publicly available at https://lammps.sandia.gov/. Datasets generated during the current study and custom codes written to analyze data are available from the corresponding author upon request.

## Acknowledgments

We are grateful to C.A. Brackley, S. Franzini, C. Micheletti, and D. Michieletto for useful discussions.

We thank the Wellcome Trust (223097/Z/21/Z to N. Gilbert and D. Marenduzzo) and the European Research Council (CoG 648050 THREEDCELLPHYSICS to D. Marenduzzo) for funding.

Author contributions: G. Forte, L. Boteva, N. Gilbert, P.R. Cook, and D. Marenduzzo designed research; G. Forte, F. Conforto, and D. Marenduzzo performed simulations; L. Boteva performed lab-based research; G. Forte, L. Boteva, N. Gilbert, P.R. Cook, and D. Marenduzzo analyzed the data; and G. Forte, L. Boteva, N. Gilbert, P.R. Cook, and D. Marenduzzo wrote the manuscript.

Disclosures: The authors declare no competing interests exist.

Submitted: 28 September 2022

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

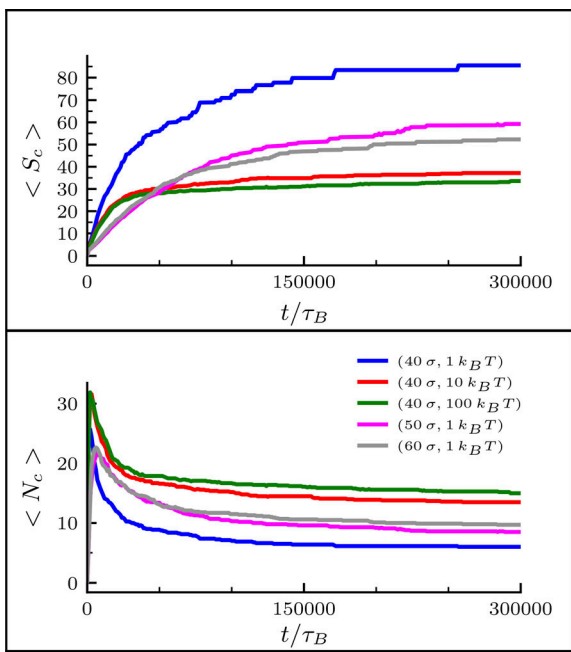

**Figure S1. Clustering analysis for bridging condensins during mitotic chromosome compaction.** Plots showing the average cluster size <$S_c$> (top panel) and the average number of clusters <$N_c$> (bottom panel) versus time during the formation of a SAC starting from a BBP configuration. The clustering analysis has been performed on condensin bridges. Different colors refer to different values of ($L_{loop}$, $A$). The color legend is displayed in the bottom panel, where σ corresponds to 2 kbp. The average is computed over 10 simulations.

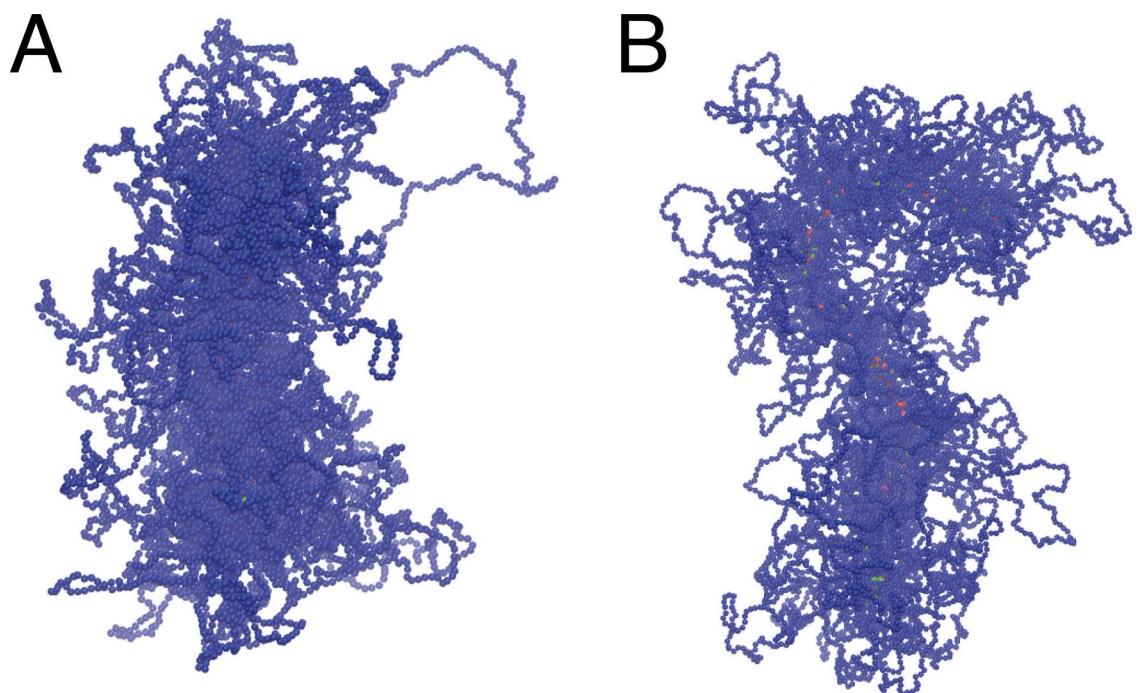

Figure S2. **Snapshots of mitotic chromosomes with Poisson-distributed loops.** Two examples of how mitotic cylinders appear when their loops are distributed accordingly to a Poisson distribution. **(A and B)** In both panels, the average loop length is $L_{loop}$ = 80 *kbp*, while the potential among polymer beads is $A$ = 1 $k_BT$ (A) and $A$ = 10 $k_BT$ (B).

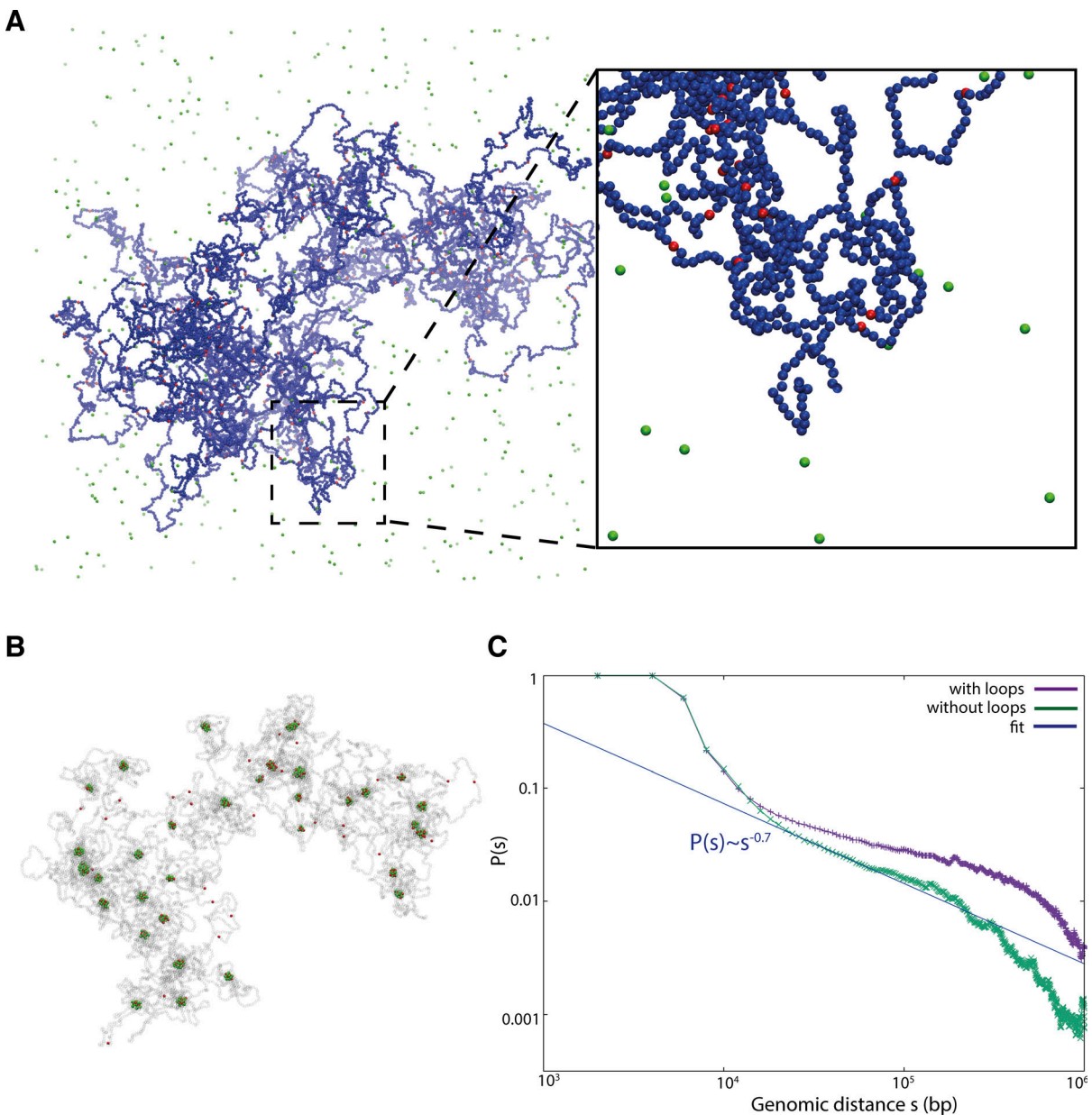

Figure S3. **Analysis of the contact probability in the absence of chromatin condensin loops. (A and B)** Snapshots from simulations where chromatin loops have been removed and condensin bridges (green beads) can tightly bind chromatin on specific sites (red beads) and weakly bind any other sites (blue beads). Panel A shows the initial configuration where chromatin and bridges sterically interact. When attractive interactions between chromatin and bridges are switched on, the system reaches a new steady-state and condensin bridges form clusters around the specific chromatin sites (panel B; blue beads are here represented as transparent). **(C)** Contact probability versus genome distance for standard simulations (with loops) and for simulations represented in panels A and B (without loops). In the case without loops, it is not possible to observe the characteristic power law P(s) ∼ s−0.5 observed in mitotic chromosomes.

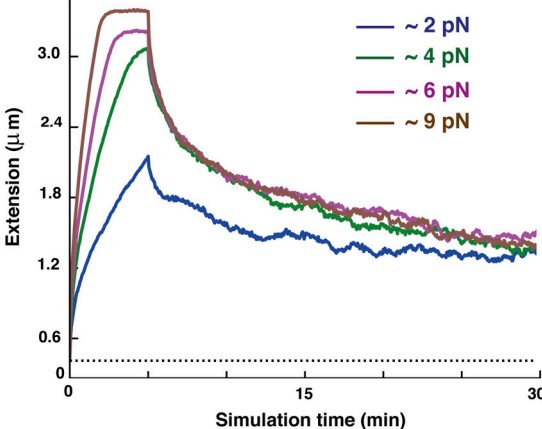

Figure S4. **Mitotic chromosome extension during an extension–retraction cycle.** Mitotic chromosome extension versus time during an extension–retraction cycle performed in the absence of bridging activity. It is possible to observe that the lack of bridging condensins prevents chromosomes from going back to the original extension (dotted horizontal line).

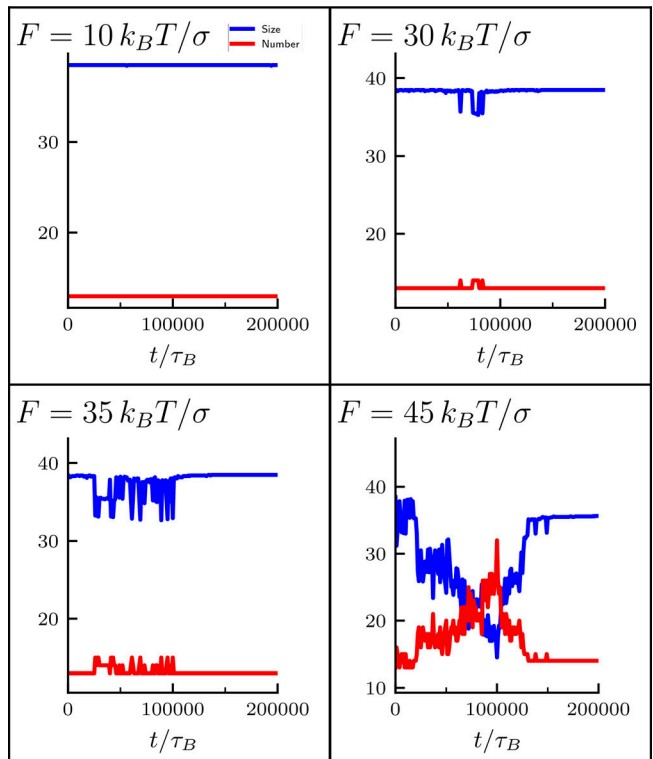

Figure S5. **Cluster analysis for condensin bridges during an extension–retraction cycle.** In the figures, the cluster size (blue curves) and the number of clusters (red curves) are displayed for a single extension–retraction cycle performed at different pulling forces (F = 2, 6, 7, and 9 pN as $k_B T/\sigma$ corresponds to 0.2 pN).

Video 1. **Self-assembling of a mitotic chromosome mediated by bridging condensins.** Trajectory showing the compaction of a BBP following the formation of condensin-mediated bridges. Initially, bridges diffuse in the simulation box, and later on an attractive interaction between them and the chromosome is switched on, resulting in the compaction of the bottle-brushed polymer in a cylindrical shape. The movie refers to a BBP with fixed loop size $L_{loop}$ = 80 kbp and soft potential A = 10 $k_B T$. Video edited at 6 fps.

**Video 2.** **Bridging-mediated compaction of bottlebrush sister chromatids.** Compaction of two bottlebrush chromatids connected through the centromere represented by using additional springs. The folding is driven by the attractive interaction between the polymers and bridging condensins. The two chromatids are composed by loops of fixed size $L_{loop}$ = 80 $kbp$ and soft potential $A$ = 10 $k_BT$. Video edited at 20 fps.

**Video 3.** **Bridging-mediated folded chromosomes are elastic objects.** The two extremities of a self-assembled mitotic chromosome are pulled with a constant force $F$ = 6 $pN$. The chromosome is characterized by a fixed loop length $L_{loop}$ = 80 $kbp$ and soft potential $A$ = 10 $k_BT$. After an extension phase, when the polymer is stretched, the pulling force is switched off and the chromosome is allowed to relax and its length gets close again to its original value. Video edited at 4 fps.

