## [Peer Review File · The Journal of Cell Biology]

Bridging condensins mediate compaction of mitotic chromosomes

Giada Forte, Lora Boteva, Filippo Conforto, Nick Gilbert, Peter Cook, and Davide Marenduzzo

Corresponding Author(s): Davide Marenduzzo, University of Edinburgh and Giada Forte, University of Edinburgh

Review Timeline:

Submission Date:	2022-09-28
Editorial Decision:	2023-03-11
Revision Received:	2023-07-08
Editorial Decision:	2023-07-18
Revision Received:	2023-08-03

Monitoring Editor: Arshad Desai

Scientific Editor: Dan Simon

Transaction Report:

DOI: <https://doi.org/10.1083/jcb.202209113>

March 11, 2023

Re: JCB manuscript #202209113

Prof. Davide Marenduzzo
University of Edinburgh
Peter Guthrie Tait Road
Edinburgh EH9 3FD
United Kingdom

Dear Prof. Marenduzzo,

Thank you for submitting your manuscript entitled "Bridging-mediated compaction of mitotic chromosomes." Your manuscript has been assessed by 3 expert reviewers, whose comments are appended to this letter. Thank you for your patience with the peer review process. Although the reviewers express potential interest in this work, significant concerns unfortunately preclude publication of the current version of the manuscript in JCB.

The reviewers all appreciate the model presented in your work but, while Reviewers 1 and 2 are supportive, Reviewer 3 is critical of the current submission. After considering the reviewers comments editorially, our view is that for the manuscript to be suitable for the journal a thorough revision addressing the different points raised by the reviewers is essential and any submitted revision will need to undergo re-review. If you decide to pursue this course of action, our recommendation is to place the work better in the context of the literature and take significantly greater care in distinguishing hypotheses/ideas from experimentally-established views as well as extend the work, when feasible, along the lines recommended by Reviewer 3. We also would like to emphasize that the figures and text should be overhauled to appeal to the broad cell biology audience targeted by the journal. We recommend for both of these points to follow the recommendations made by the reviewers.

Please let us know if you are able to address the major issues outlined above and wish to submit a revised manuscript to JCB. The typical timeframe for revisions is three to four months. While most universities and institutes have reopened labs and allowed researchers to begin working at nearly pre-pandemic levels, we at JCB realize that the lingering effects of the COVID-19 pandemic may still be impacting some aspects of your work, including the acquisition of equipment and reagents. Therefore, if you anticipate any difficulties in meeting this aforementioned revision time limit, please contact us and we can work with you to find an appropriate time frame for resubmission. Please note that papers are generally considered through only one revision cycle, so any revised manuscript will likely be either accepted or rejected.

If you choose to revise and resubmit your manuscript, please also attend to the following editorial points. Please direct any editorial questions to the journal office.

GENERAL GUIDELINES:

Text limits: Character count is < 40,000, not including spaces. Count includes title page, abstract, introduction, results, discussion, and acknowledgments. Count does not include materials and methods, figure legends, references, tables, or supplemental legends.

Figures: Your manuscript may have up to 10 main text figures. To avoid delays in production, figures must be prepared according to the policies outlined in our Instructions to Authors, under Data Presentation, <https://jcb.rupress.org/site/misc/ifora.xhtml>. All figures in accepted manuscripts will be screened prior to publication.

Supplemental information: There are strict limits on the allowable amount of supplemental data. Your manuscript may have up to 5 supplemental figures. Up to 10 supplemental videos or flash animations are allowed. A summary of all supplemental material should appear at the end of the Materials and methods section.

Please note that JCB now requires authors to submit Source Data used to generate figures containing gels and Western blots with all revised manuscripts. This Source Data consists of fully uncropped and unprocessed images for each gel/blot displayed in the main and supplemental figures. File names for Source Data figures should be alphanumeric without any spaces or special characters (i.e., SourceDataF#, where F# refers to the associated main figure number or SourceDataFS# for those associated with Supplementary figures). The lanes of the gels/blots should be labeled as they are in the associated figure, the place where cropping was applied should be marked (with a box), and molecular weight/size standards should be labeled wherever possible. Source Data files will be made available to reviewers during evaluation of revised manuscripts and, if your paper is eventually

published in JCB, the files will be directly linked to specific figures in the published article.

If you choose to resubmit, please include a cover letter addressing the reviewers' comments point by point. Please also highlight all changes in the text of the manuscript.

Regardless of how you choose to proceed, we hope that the comments below will prove constructive as your work progresses. We would be happy to discuss them further once you've had a chance to consider the points raised. You can contact the journal office with any questions, cellbio@rockefeller.edu or call (212) 327-8588.

Thank you for thinking of JCB as an appropriate place to publish your work.

Sincerely,

Arshad Desai, PhD
Monitoring Editor
Journal of Cell Biology

Dan Simon, PhD
Scientific Editor
Journal of Cell Biology

Reviewer #1 (Comments to the Authors (Required)):

The manuscript studies, by means of computer simulations, a coarse-grained polymer model which addresses the compaction of mitotic chromosomes. When leaving the interphase the chromosomes become shorter and condense into a cylindrical shape. As explained in the manuscript, polymer physics would favor a condensation into spherical globules. The authors suggest that condensin I acts as a bridge and preferentially attach to the loop anchors of a relaxed bottlebrush chromosome. By doing so they form clusters that stiffen and condense the chromosomes. The model, although rather simple, is capable to capture some experimentally observed features.

I enjoyed reading the manuscript and I am of the opinion that this work deserves to be published. It contains a detailed simulation study of a model of chromosome condensation that reports quantitative data which can inspire further experiments. I have just a few remarks that the authors should take into account in a revision round.

I think that the authors should be clearer in the introduction about what is known from the biology of condensin chromosome interactions and what is speculation. I was confused about the description of two modes of interactions of condensins. When the authors refer to the bridging of two different genomic segments they mention they cite Ref.10, which is about cohesin, right? Are there any similar observations of bridging for condensin?

In the discussion they mention again "...experiments showing evidence for both looping and bridging activities of condensins and related SMC proteins (10;37)". Actually also Ref.(37) is about cohesin. In addition the interaction modes cohesin discussed in Ref. (37) imply embracement of DNA, while in the introduction it is mentioned a bridging without embracement.

It is fine with me if the authors assume a bridging interaction of condensin, speculating that it should be similar to that observed for cohesin, but this should be more explicitly mentioned in the manuscript. I suggest the author rework the introduction adding more of these details.

I am confused about the coarse-graining distances. How can a bead of 10-30nm of diameter contain 2kbp of DNA?

There is a typo on pag.5: "..opposite forces +-F erre applied..." which should be "..were applied..."

Reviewer #2 (Comments to the Authors (Required)):

In the manuscript "Bridging-mediated compaction of mitotic chromosomes," Forte and coauthors investigate the process of mitotic compaction and propose a simplified model about how cohesin and condensin (two protein complexes of the SMC family)

might act in synergy to shape cylindrical chromosomes. Results from their model seem in reasonable agreement with experimental observations.

The arguments presented, the supporting simulations, and the clarity of text are all good and, in my opinion, make the manuscript worthy of publication. However, there also exist some issues that need to be resolved before I can fully support publication, which I discuss below.

The first is that many hypotheses seem to be presented as facts. The existence of a backbone in mitotic chromosomes composed of condensin, its alleged helical shape, and the bottlebrush nature of mitotic chromosomes are all hypotheses that have been suggested in the literature but that are far from being generally accepted. These very plausible hypotheses often agree with experimental observations but so do others. The very model presented in this manuscript seems to agree with observations as well, yet, for the moment also remains an interesting hypothesis. The problem partially lies in the fuzziness of experimental observations, that cannot yet truly discriminate among the different mechanisms that are proposed. Because of that, is paramount for every new manuscript to be meticulous in distinguishing between established facts and mere hypotheses and in presenting all points of view present in the literature. The manuscript should be revised, weakening these statements and broadening the point of view.

The second issue is an incomplete discussion of previous results. In particular, there are a few very relevant recently published results that should be discussed in the context of the model proposed by Forte and coauthors. Hoencamp et al (<https://doi.org/10.1126/science.abe2218>) discusses the result of condensin II knockout and rationalizes these results with a theoretical model. Brahmachari et al (<https://doi.org/10.1093/nar/gkac231>) also elaborates on the effects of condensin and cohesin activity in determining the geometrical shape of chromosomes. Contessoto et al (<https://doi.org/10.1038/s41467-023-35909-2>) presents an interesting analysis on the helicity of chromosomes.

Reviewer #3 (Comments to the Authors (Required)):

Forte et al., perform simulations of the DNA compaction using the bridging model. Understanding the mechanisms of DNA compaction is one of the outstanding challenges in the field and thus the topic the research is interesting. In this work, authors make model assumptions that don't seem to be based on the molecular data, and the comparison between the proposed and the other possible models is missing. Thus, at this stage the manuscript cannot establish whether the bridging must be involved in the chromosome compaction or generally what the bridging can or cannot do.

Major points

The possibility of the condensin-condensin interactions has been suggested before, but no direct evidence exists so far. In single molecule assays, condensin does not show tendency to oligomerize. Consistent with this, loop-extrusion assays show that a single condensin is responsible for generating DNA loops and thus there is no evidence of bridging interactions by condensins.

The model seems to make somewhat arbitrary assumptions about the nature of the condensin-condensin and condensin-DNA interactions: "Condensin bridges are modelled as diffusing beads (shown in green in Fig. 1) that bind reversibly and weakly to chromatin blue beads and strongly to red beads - the loop anchors". Thus, the model assumes that the interaction between the individual condensins is stronger than the interaction between condensins and DNA. If that was the case, all condensin molecules would be in oligomers before they get a chance to bind DNA. There is no molecular evidence to back this assumption.

Authors propose that both types of condensins should have both types of activities, loop extrusion and bridging. This cannot be ruled out, but this assumption seems to make things more confusing rather than clearing them up. It has been shown that loop-extrusion alone can explain chromosome compaction in chicken cells and the bridging can explain compaction in yeast. Combining the two models is likely to give even better fit to the data. But how would this give new insights into the mechanisms of compaction? Authors should show the consequences of having either of these activities separately and what the benefits of their joint action would be.

Can a single activity of condensin explain observed compaction of the chromosomes? Can bridging alone explain the Hi-C data? If not, what is it that bridging can do that loop extrusion cannot? Can loop-extrusion explain chromosome stretching data, or is the bridging required?

Figure 2C(ii) is the only data that compares the results of the simulations with experimental data. It is presented in such a way that it is difficult to see (different graphs and different axes), so it is easy to make a mistake. However, there seems to be little or no match. The conclusion must be that the proposed model does not explain the existing Hi-C data.

The model does not seem to make any quantitative testable predictions. Comparisons with experiments in Figure 5 all relate to

the loop extruding activity of condensins, not the bridging.

More specific comments

First paragraph in the second results section says: "competition between the bridging-induced compaction and looping-induced stiffening of the BPP drives self-assembly into cylinders". It is unclear what it means. What is looping-induced stiffening and how does it work? Collapse of a polymer into a spherical globule should occur regardless of the stiffness of the polymer as long as its length is significantly larger than the persistence length. This must be the case even if the looped polymer is much stiffer than the naked DNA with the persistence length of just 50 nm. Why is it then that the bridging does not collapse the looped polymer into a globule?

Introduction, second line of the second paragraph reads: "by topologically loading onto fibres to stabilise long-lived loops". It was proposed that the loops are stabilized by pseudotopological loading, but there is no data so far to support the topological loading during loop extrusion.

Section "Elasticity of self-assembled cylinders mirrors that of mitotic chromosomes", end of the second paragraph it is stated that in "experiments many are pulled simultaneously". Which experiments are being discussed? In the J. Marko experiments the force was applied to a single chromosome, so a single fibre.

Next section discusses that both condensin I and II should both have loop extrusion and bridging activities. Can experimental data be explained by only one type of activity?

Other comments

Figure 1 is confusing. If blue is chromatin fiber, why is it not looped as written on line 3, first paragraph, results section? What are the grey arcs then? What does it mean "grey arcs, modelled as springs". What do springs refer to?

Figure 1. How was the experimental image of the condensin obtained? If it is from a different work, from which one?

It is difficult to read the panels with four gradations of levels that are referred to as Figure 2, panel C, subpanel (ii), image (top), (left), (bottom), (center), etc. It would be very helpful if each panel were referred to by its own index. 2a, 2b, etc.

Figure 2B. What are the different color curves? What model parameters correspond to the each color?

Authors should include the table of model parameters. For each parameter, the range of values should be indicated, and it should be clear when parameters are estimated from experiments and when they are variable.

Figure 2C(ii). The comparison between the experimental and theoretical curves should be presented on the same graph with the same axes. As it is shown here, it is almost impossible to compare the curves.

For the publication in the Journal of Cell Biology, physical quantities in figures should be given in physical units. Time should be in seconds (or minutes, etc, but not relative simulation time), extension should be given in nanometers, microns, etc. Cases where it is not possible should be explained. Model parameters used to simulate curves should be clearly indicated on the figure.

Dear Editor,

Many thanks for sending us your decision and the Reviewers' comments on our manuscript, "Bridging-mediated compaction of mitotic chromosomes".

As discussed and planned in our previous correspondence, we have now revised the manuscript according to your and the Reviewers' suggestions, and also performed a number of additional simulations to address the constructive criticism provided by the Reviewers. In particular we provide a number of results on the behaviour of a system with only the condensin bridging or only the condensin looping activity. Additional data showing new results are shown in the new Figures 5, 7, S3 and S4.

Below are a reply to Editorial comments and a point-by-point reply to the Reviewers' comments and questions. Note that the main changes in the manuscript and Supplemental Information are highlighted in blue. We hope that the revised manuscript can now be published in J. Cell Biol.

With best wishes,

G. Forte, L. Boteva, F. Conforto, N. Gilbert, P. R. Cook, D. Marenduzzo

Reply to Editorial comments

The reviewers all appreciate the model presented in your work but, while Reviewers 1 and 2 are supportive, Reviewer 3 is critical of the current submission. After considering the reviewers comments editorially, our view is that for the manuscript to be suitable for the journal a thorough revision addressing the different points raised by the reviewers is essential and any submitted revision will need to undergo re-review. If you decide to pursue this course of action, our recommendation is to place the work better in the context of the literature and take significantly greater care in distinguishing hypotheses/ideas from experimentally-established views as well as extend the work, when feasible, along the lines recommended by Reviewer 3. We also would like to emphasize that the figures and text should be overhauled to appeal to the broad cell biology audience targeted by the journal. We recommend for both of these points to follow the recommendations made by the reviewers.

Many thanks for your comments. We reply to all Editorial points below.

First, we appreciated the suggestions to place the work better in the context of the literature and to take care in distinguishing hypotheses from experimentally-established views. To address this point, we have now enlarged the discussion of related work in the Introduction and Discussion, including all the relevant previous works highlighted by the Reviewers. We also took particular care in highlighting all points in our model where hypothesis/ideas were introduced, and in discussing the underlying experimental evidence.

Second, we have performed a number of additional simulations to clarify the distinct roles of bridging and looping by condensins, with regards both to the formation of mitotic chromosomes and to the chromatin structure around common fragile sites.

Third, we have now substantially changed the Figure presentation to address the comments by the Reviewers and to appeal the broad cell biology audience of the journal.

We have also taken into account also all additional Editorial points, and confirm that we adhere to the journal guidelines.

Below we provide a point-by-point response to each of the Reviewers' comments.

Point-by-point response to Reviewers' comments

In what follows, each of the Reviewers' questions is highlighted in blue, while our response is in larger black text.

Reviewer #1

The manuscript studies, by means of computer simulations, a coarse-grained polymer model which addresses the compaction of mitotic chromosomes. When leaving the interphase the chromosomes become shorter and condense into a cylindrical shape. As explained in the manuscript, polymer physics would favor a condensation into spherical globules. The authors suggest that condensin I acts as a bridge and preferentially attach to the loop anchors of a relaxed bottlebrush chromosome. By doing so they form clusters that stiffen and condense the chromosomes. The model, although rather simple, is capable to capture some experimentally observed features.

I enjoyed reading the manuscript and I am of the opinion that this work deserves to be published. It contains a detailed simulation study of a model of chromosome condensation that reports quantitative data which can inspire further experiments. I have just a few remarks that the authors should take into account in a revision round.

We thank the reviewer for carefully reading our work and for the positive feedback provided. Below we reply to their comments and questions, which we feel have led us to improve the manuscript.

I think that the authors should be clearer in the introduction about what is known from the biology of condensin chromosome interactions and what is speculation. I was confused about the description of two modes of interactions of condensins. When the authors refer to the bridging of two different genomic segments they mention they cite Ref.10, which is about cohesin, right? Are there any similar observations of bridging for condensin?

In the discussion they mention again "...experiments showing evidence for both looping and bridging activities of condensins and related SMC proteins (10;37)". Actually also Ref.(37) is about cohesin. In addition the interaction modes cohesin discussed in Ref. (37) imply embracement of DNA, while in the introduction it is mentioned a bridging without embracement.

It is fine with me if the authors assume a bridging interaction of condensin, speculating that it should be similar to that observed for cohesin, but this should be more explicitly mentioned in the manuscript. I suggest the author rework the introduction adding more of these details.

First, we thank the reviewer for highlighting a potentially confusing point of our paper. As they said, we introduce two types of condensins (or condensing activities) in our model. The first one creates stable and long-lived chromatin loops, for instance through the loop extrusion mechanism. The second one mediates chromatin bridges between far chromatin sites by multivalent binding the chromatin fibre. In the revised version of the paper we refer to the first condensin as "looping condensin", and to the second one as "bridging condensin".

While the ability of condensins to create chromatin loops has been observed in in-vitro experiments (Ganji et al, Science, 2018), the bridging activity for condensing is more speculative, and is based on work on other SMC complexes, as mentioned by the Reviewer, and we have now highlighted this. More specifically, as the reviewer observed, Ref (37) and Ref(10) show that other SMC complexes, specifically cohesins, are able to create chromatin bridges. We then assume that this bridging mechanism applies to condensins too as they are SMC complexes like cohesins.

More generally, in the new version of our paper, we have now underlined what is supported by experiments and previous results and what instead is a new assumption we do.

Additionally, as also discussed in the reply to Reviewer 3 below, in the original version of the manuscript, we used the terms *topological* and *non-topological* to refer to looping and bridging condensins respectively. In the revised version we removed these terms as we realised they can generate confusion when compared to the results of Ref(37). As we are performing coarse-grained simulations, we are not including details about the topological nature condensin loading on the chromatin fibre. For this reason, we do not need to specify whether our condensin loading implies embracement or not, but we simply mention looping and bridging activities of condensins.

I am confused about the coarse-graining distances. How can a bead of 10-30nm of diameter contain 2kbp of DNA?

We have now specified that we assume a chromatin bead to contain 2 kbp, leading to a diameter of about 20nm (see, e.g., J. Langowski, The European Physical Journal E, 2006).

Reviewer #2

In the manuscript "Bridging-mediated compaction of mitotic chromosomes," Forte and coauthors investigate the process of mitotic compaction and propose a simplified model about how cohesin and condensin (two protein complexes of the SMC family) might act in synergy to shape cylindrical chromosomes. Results from their model seem in reasonable agreement with experimental observations.

The arguments presented, the supporting simulations, and the clarity of text are all good and, in my opinion, make the manuscript worthy of publication. However, there also exist some issues that need to be resolved before I can fully support publication, which I discuss below.

We would like to thank the reviewer for reading our work and for recommending it for publication subject to the resolution of the issues mentioned. Below we address the Reviewer's comments in detail.

The first is that many hypotheses seem to be presented as facts. The existence of a backbone in mitotic chromosomes composed of condensin, its alleged helical shape, and the bottlebrush nature of mitotic chromosomes are all hypotheses that have been suggested in the literature but that are far from being generally accepted. These very plausible hypotheses often agree with experimental observations but so do others. The very model presented in this manuscript seems to agree with observations as well, yet, for the moment also remains an interesting hypothesis. The problem partially lies in the fuzziness of experimental observations, that cannot yet truly discriminate among the different mechanisms that are proposed. Because of that, it is paramount for every new manuscript to be meticulous in distinguishing between established facts and mere hypotheses and in presenting all points of view present in the literature. The manuscript should be revised, weakening these statements and broadening the point of view.

We agree with the Reviewer that it is important to clearly specify the hypotheses and assumptions of our model. In the revised version of the manuscript we have now done this. Specifically, we have provided more details about previous literature and specified what facts we base our model on and where we make (reasonable) assumptions. In particular the presence of the condensin backbone, the bottlebrush nature of the fibre, and the bridging action of condensin are now presented on

hypotheses, some of which are partially but not fully supported by existing experimental knowledge.

The second issue is an incomplete discussion of previous results. In particular, there are a few very relevant recently published results that should be discussed in the context of the model proposed by Forte and coauthors. Hoencamp et al (<https://doi.org/10.1126/science.abe2218>) discusses the result of condensin II knockout and rationalizes these results with a theoretical model. Brahmachari et al (<https://doi.org/10.1093/nar/gkac231>) also elaborates on the effects of condensin and cohesin activity in determining the geometrical shape of chromosomes. Contessoto et al (<https://doi.org/10.1038/s41467-023-35909-2>) presents an interesting analysis on the helicity of chromosomes.

We thank the reviewer for highlighting this relevant previous. We have now included and briefly discussed the suggested references in our manuscript.

Reviewer #3

Forte et al., perform simulations of the DNA compaction using the bridging model. Understanding the mechanisms of DNA compaction is one of the outstanding challenges in the field and thus the topic the research is interesting. In this work, authors make model assumptions that don't seem to be based on the molecular data, and the comparison between the proposed and the other possible models is missing. Thus, at this stage the manuscript cannot establish whether the bridging must be involved in the chromosome compaction or generally what the bridging can or cannot do.

First, we would like to thank the reviewer for the time spent in reading our manuscript, and for providing constructive criticism. In the revised manuscript we have made several changes and performed additional simulations to support our model, and address the criticism.

Specifically, we have now made much clearer which assumptions we are doing and how we can justify them from a molecular biology point of view. Additionally, the new simulations now prove the importance of the bridging activity of condensin. In particular, in the Supplementary Information part we included:

1. Simulations modelling condensin knock-outs, and chromatin folding at common fragile sites, by modelling the removal of selected condensin activities. The new results underscore how the absence of bridging condensin activity *cannot* reproduce microscopy experiment observations.
2. Simulations including only condensin looping (e.g., due to loop extrusion) showing that this mechanism cannot provide the same compaction as the one observed by inserting condensin bridges which well captures properties of prometaphase chromosomes.

Major points

The possibility of the condensin-condensin interactions has been suggested before, but no direct evidence exists so far. In single molecule assays, condensin does not show tendency to oligomerize. Consistent with this, loop-extrusion assays show that a single condensin is responsible for generating DNA loops and thus there is no evidence of bridging interactions by condensins.

In our model we simulate the compaction of mitotic chromosomes through the action of two types of condensins, one creating stable chromatin loops (which might be obtained through loop extrusion mechanisms) and the other forming molecular bridges.

For the first point mentioned by the Reviewer, it is important to underline that the latter experiences an attraction to the chromatin fibre, but **no attraction between condensin-like bridges** is assumed or included in the model. The formation of bridge clusters is instead essentially due to the bridging-induced attraction mechanism discovered in our group (see, e.g., C. A. Brackley et al., PNAS 110, E3605 (2013)), which can be summarised as follows. When attractive interactions are inserted between specific chromatin sites and protein complexes, the concentration of binding sites increases following bridges formation. This, in turns, leads to more protein complexes binding the chromatin filament and to the formation of clusters (see Fig. Rep1).

Figure Rep1: The introduction of specific interaction between the chromatin filament (grey line) and protein complexes (red beads) results in the increase of the local concentration of chromatin binding sites which provokes the binding of additional proteins and the formation of protein clusters. Figure adapted from Chiang et al, Hi-C Data Analysis: Methods and Protocols, 267-291, 2022.

For the assumption of bridging condensin activity, we have now highlighted the fact that this is an assumption, which remains speculative but is reasonable in view of the documented multivalent bridging (or cross-linking) activity of other SMC complexes such as cohesins. We note that other groups have hypothesised such an activity (this is directly hypothesised in T. M. K. Cheng et al., eLife 05565 (2015), T. Gerguri et al., NAR 49, 1294 (2021), and consistent with the hypothesised non-extrusion condensin activity in K. Kinoshita, J. Cell. Biol. 221, e202109016 (2022)). We have now cited these supporting studies which make similar hypothesis to ours, which are partially but not yet fully based on experimental evidence.

The model seems to make somewhat arbitrary assumptions about the nature of the condensin-condensin and condensin-DNA interactions: "Condensin bridges are modelled as diffusing beads (shown in green in Fig. 1) that bind reversibly and weakly to chromatin blue beads and strongly to red beads - the loop anchors". Thus, the model assumes that the interaction between the individual condensins is stronger than the interaction between condensins and DNA. If that was the case, all condensin molecules would be in oligomers before they get a chance to bind DNA. There is no molecular evidence to back this assumption.

As explained in the previous point, our model does not introduce interactions among bridging condensins, hence excluding they form oligomers before binding DNA. We trust this is now clear, and apologise that this might have been unclear in the original version.

Regarding the remaining part of this comment, we indeed model bridging condensins as proteins interacting with chromatin through two different potentials: a weak attractive potential with non-specific sites (blue beads in Fig.1) and a strong attractive potential with loop roots (red beads in Fig.1) which correspond also to the position of what we refer as looping condensins, i.e. condensins acting as molecular motors and extruding a loop. Nonetheless, the stronger interaction between bridging condensins and loop roots does not imply a stronger condensin-condensin interaction compared to condensin-DNA interaction. Immunofluorescent experiments revealed that both condensin types involved in mitosis (condensin I and II) are located along the

chromosome axis (Takao Ono et al, Cell, 115, 109-121 (2003)) (Green et al, Journal of cell science, 125, 1591-1604 (2012)) suggesting that both these SMC complexes might have a high affinity with the same specific chromatin sites. As our bridging and looping condensin activities model both condensin I and II (as it is reasonable to assume that both would have similar activities), it is also reasonable to assume that they both bind the same chromatin sites. This justifies the higher affinity of bridging condensins with chromatin loop roots without requiring an attractive interaction between them and looping condensins. We have highlighted though that this is an assumption, and clarified the molecular basis according to which we feel this is reasonable.

Authors propose that both types of condensins should have both types of activities, loop extrusion and bridging. This cannot be ruled out, but this assumption seems to make things more confusing rather than clearing them up. It has been shown that loop-extrusion alone can explain chromosome compaction in chicken cells and the bridging can explain compaction in yeast. Combining the two models is likely to give even better fit to the data. But how would this give new insights into the mechanisms of compaction? Authors should show the consequences of having either of these activities separately and what the benefits of their joint action would be.

Can a single activity of condensin explain observed compaction of the chromosomes? Can bridging alone explain the Hi-C data? If not, what is it that bridging can do that loop extrusion cannot? Can loop-extrusion explain chromosome stretching data, or is the bridging required?

This is an important and valid question, and has prompted us to do additional simulations to show more quantitatively the key importance of having both activities in our model.

More specifically, in our model we propose the joint action of bridging condensins and loop extruding condensins is required to explain mitotic chromosome folding after prophase. Interestingly, as we show in the main text, the interplay of the two mechanisms (looping and bridging) provides several results in agreement with experimental data. On the other hand, it is indeed interesting to consider the two mechanisms in isolation. As we show more clearly with our additional simulations, each mechanism alone can explain only some of the experimental, but also leave open questions as we explain in more detail below. Indeed, the previous studies using a single mechanisms mentioned by the Reviewer need to make additional assumptions to make qualitative contact with the experiments.

Looping and loop extrusion

Hi-C and 5C data were used to investigate the folding of human mitotic chromosomes (Naumova, Science, 2013). The inferred contact probability vs genomic distance curve for metaphase chromosomes was employed to fit several polymer models revealing that the most likely mitotic structure corresponds to a bottle-brush polymer composed by consecutive loops with a size between 80 kbp and 120 kbp. Further microscopy and chromosome conformation capture experiments performed on chicken DT-40 cells were used to develop other polymer models which shed light on the different roles played by condensin I and II during mitotic compaction (Gibcus, Science, 2018). Importantly, polymer models employed in both these works, require additional axial compression of the bottle-brush polymer. This compression is obtained, for instance, by confining the bottle-brush polymer inside a cylinder (Gibcus, Science, 2018). In

other words, the looping activity by itself **is not** sufficient to give a cylindrical shape to chromosomes in the mentioned work on chicken cells.

Additionally, in previous theoretical work studying chromatin compaction by loop extrusion alone (Goloborodko, eLife, 2016), authors clearly specify that their simulations can reproduce mitotic compaction up to prophase, but they cannot capture the further compaction into prometaphase and metaphase cylinders.

Finally, we highlight that the loop extrusion model alone could not reproduce chromosome stretching data obtained from single-molecule experiments. To prove this, we have performed new simulations where we pull the two extremities of a bottle-brush polymer used in our model, but without introducing condensin-like bridges. In this case, as in the main text, loops are formed by introducing a harmonic potential between consecutive loop roots. By applying the same pulling forces used in our main text, we see that the absence of bridges would prevent chromosomes to go back to the original extension after being stretched (see figure Rep2). Bridging activity therefore appears to be fundamental to reproduce mitotic chromosome elasticity observed experimentally.

Figure Rep2: mitotic chromosome extension versus time during an extension-retraction cycle performed in absence of bridging activity. It is possible to observe that the lack of condensin bridges prevents chromosomes to go back to the original extension (dotted horizontal line).

Bridging

As the reviewer observed, an alternative approach consists in explaining mitotic compaction through the action of condensins acting as bridges. This method was used by Cheng et al (Cheng, eLife, 2015) to investigate mitotic folding in yeast cells. While

simulated 4C profiles seem to yield good agreement with experimental data, the authors already clearly acknowledged that that model is an oversimplification as, for instance, it does not include the two condensin types which are found in higher eukaryotes.

To prove that bridging alone could not provide good agreement with metaphase Hi-C experimental data, we performed some additional molecular dynamics simulations. The contour length of the polymer is kept fixed as well as the number of condensin-like bridges. As in the model by Cheng et al, we now consider a linear polymer (i.e. without chromatin loops, see figure below) which is composed by non-specific sites (blue beads) and specific sites (red beads). Blue and red beads have a weak and strong attractive interaction with condensin-like bridges (green beads) respectively, leading to the formation of condensin clusters in proximity of the chromatin red beads (see figure below). The number of red beads in this new model is in line with the number of loop roots in our original model.

By performing a set of 10 uncorrelated simulations, we then compute the contact probability as a function of the genome distance. As we see from Fig. Rep3, it is not possible to reproduce the characteristic exponential law $P(s) \propto s^{-0.5}$ observed from Hi-C experiments. Instead, the contact exponent appears now to be -0.7 . Additionally, it is clear that the emerging shape is not cylindrical.

Figure Rep3. (A-B): Snapshots from simulations where looping condensins have been removed and only bridging condensins in modelled. Condensin bridges (green beads) can tightly bind chromatin on specific sites (red beads) and weakly bind any other sites (blue beads). Panel (A) shows the initial configuration where chromatin and bridges sterically interact. When attractive interactions between chromatin and bridges are switched on, the system reaches a new steady state and condensin bridges form clusters around the specific chromatin sites (panel (B)) -- blue beads

are here represented as transparent). **(C)** Contact probability versus genome distance for standard simulations (with loops) and for simulations represented in panels (A-B) (without loops). In the case without loops it is not possible to observe the characteristic power law $P(s) \sim s^{-0.5}$ observed in mitotic chromosomes, but we see instead $P(s) \sim s^{-0.7}$. Additionally, it can be seen that the emerging shape of the polymer is not cylindrical.

From the considerations above, it can be seen that both the extruding/looping and bridging activity in isolation are not enough to qualitatively agree with experiments, whereas the combination of these, as discussed in the main text, lead to: (i) cylindrical shape, (ii) Hi-C contact decay, and (iii) mitotic chromosome elasticity under stretching experiments. A single activity does not recapitulate (i) in metaphase, and fails to capture (ii) and (iii) as well.

In particular, the combination of activity is important to model compaction after prophase. Of course, with our modelling what we can do is only to suggest an alternative scenario which needs to be confirmed through experiments, and we are also simplifying the system by inserting just a few elements involved in mitosis (i.e. condensins and topoisomerases).

We have now added the results obtained with simulations considering a single condensin activity in the Supplementary Information, discussing in the manuscript the results, which highlight the advantage of having both condensin activities. We thank the Reviewer for prompting us to do more work to clarify the importance of the combination of the two activities.

Figure 2C(ii) is the only data that compares the results of the simulations with experimental data. It is presented in such a way that it is difficult to see (different graphs and different axes), so it is easy to make a mistake. However, there seems to be little or no match. The conclusion must be that the proposed model does not explain the existing Hi-C data.

We have now updated the previous Fig. 2C(ii) by using physical units also for our simulation results and by combining the latter with experimental data from Naumova et al, Science, 2013. It is now clearer how the two sets of data (simulated and experimental) follow the same power law, even if in simulations the genomic distance where the power law holds is smaller. As explained in the manuscript, this discrepancy is likely due to the fact that our simulated chromosomes are shorter than human chromosomes investigated in Naumova et al, and that the data shown are for late time, during metaphase, where additional compaction is likely to occur *in vivo* (e.g., associated with the formation of a helix hypothesised in Gibcus et al., Science, 2018, which is not captured by our model), which would extend the genomic distance range where the power law remains valid.

The model does not seem to make any quantitative testable predictions. Comparisons with experiments in Figure 5 all relate to the loop extruding activity of condensins, not the bridging.

Figure 5 (now Fig. 6, redrawn, as all Figures, for added clarity, see below) has the aim of reproducing local defects visible at common fragile sites through molecular dynamics simulations. The two types of defects examined here are cytological lesions and irregular FISH phenotypes. These are actually associated with defects in both looping and bridging activities, as we hope transpires more clearly from the revised text. Briefly, both defects are simulated through a faulty loading of both looping and bridging condensins. First, in the cytological lesion case both looping and bridging condensins are missing (not loaded) on part of the chromatin filament, and this leads

to a break in the backbone compatible with a cytological lesion. Second, instead, the defect referred as irregular FISH phenotype is reproduced through a reduced/faulty loading of mainly looping condensing. This results in a larger loop size which indirectly leads to local recruitment of less bridging condensins.

More specific comments

First paragraph in the second results section says: "competition between the bridging-induced compaction and looping-induced stiffening of the BPP drives self-assembly into cylinders". It is unclear what it means. What is looping-induced stiffening and how does it work? Collapse of a polymer into a spherical globule should occur regardless of the stiffness of the polymer as long as its length is significantly larger than the persistence length. This must be the case even if the looped polymer is much stiffer than the naked DNA with the persistence length of just 50 nm. Why is it then that the bridging does not collapse the looped polymer into a globule?

In our model we use a bottle-brush polymer which is composed by an array of consecutive loops. The resulting stiffness can be comparable with the whole contour length, and is determined by loop size, so it is very different from the background stiffness of the chromatin fibre (in our case this is actually a relatively flexible polymer). Therefore the picture is qualitatively different from that of a semiflexible polymer collapse. The statistical physics of mitotic and meiotic bottlebrush chromosomes has been clarified by Marko and Siggia (Marko and Siggia, *Molecular biology of the cell*, 1997) who used pre-existing results about bottle-brush polymer models (Li and Witten, *Macromolecules*, 1994) and showed that looping-induced stiffening is a massive effect.

In our model we also introduced condensin bridges. As the reviewer said, bridging mechanism alone would provoke the compaction of a linear polymer. However, as mentioned above the bottle-brush structure provides sufficient stiffness to compete thermodynamically against the formation of a globular structure. We also note that in our case there is the presence of specific binding sites (loop roots) and non-specific binding. A linear polymer composed by a few monomers with higher affinity with binders would compact into multiple clusters instead of forming a single big cluster (Chiang et al, Hi-C data analysis, 2022, and Fig. Rep3).

Of course, in the case in which the number of bridging condensins were comparable to or larger than the number of monomers composing the chromatin filament, the system would be oversaturated with bridges and the polymer would eventually collapse into a globule-like structure for sufficiently strong protein-chromatin interactions. However, this scenario is unrealistic as the required number of condensins would vastly exceed the concentration which is available in the cell.

Introduction, second line of the second paragraph reads: "by topologically loading onto fibres to stabilise long-lived loops". It was proposed that the loops are stabilized by pseudotopological loading, but there is no data so far to support the topological loading during loop extrusion.

As the Reviewer observed, in the original version of the manuscript we distinguished between topological and non-topological condensins assuming that the first perform loop extrusion, while the second ones perform bridging activity. In our coarse-grained model, there is no need to make an assumption about the topological or non-topological nature of chromatin-condensin interaction. Hence, to avoid misunderstandings on the nature of the assumptions in our model, we have reverted

to the clearer distinction between looping and bridging condensins, which more accurately reflects what done in the simulations.

Section "Elasticity of self-assembled cylinders mirrors that of mitotic chromosomes", end of the second paragraph it is stated that in "experiments many are pulled simultaneously". Which experiments are being discussed? In the J. Marko experiments the force was applied to a single chromosome, so a single fibre.

To prove that the range of pulling forces used in our simulations is realistic, we used the results from Marko's lab's experiments reported in J. F. Marko, Chromosome research, 2008. Here the author explains their results from micromanipulation experiments on newt mitotic chromosomes. The typical pulling forces applied to a whole chromosome are of the order of 1 nN, but the author also explicitly states that the cross sectional area of a newt chromosome is $\sim 3 \mu m^2$ and that *it contains thousands 30 nm chromatin fibres*. The author goes on to infer that the pulling force applied on a single fibre reduces to $\sim 1 pN$, fully in line with the ones we apply in our model.

In our simulations we model the compaction of a $\sim 30 nm$ chromatin fibre and, in the simulated set-up to reproduce elasticity experiments, the two extremities of the fibre are pulled in opposite direction with a constant force. This is therefore different from the experimental set-up where thousands of fibres are pulled together. In conclusion, we can say that the simulated pulling forces employed to extend a single fibre (between 1 pN and 5 pN) are close to the ones used in the experiments to stretch thousands of fibres together.

Next section discusses that both condensin I and II should both have loop extrusion and bridging activities. Can experimental data be explained by only one type of activity?

The section highlighted by the Reviewer is the one entitled "Simulating global condensin knock-outs, and local chromatin structure at common fragile sites", which aims to reproduce experimental results observed in condensin knock-out and in correspondence of common fragile sites. As previously discussed, we have now showed that by itself only the bridging activity could not reproduce the experimental results as local clusters would form (see Fig. Rep2) and the polymer would not assume the cylindrical shape typical of mitotic chromosomes. Therefore, the questions asked in this Section cannot be addressed with only a bridging activity. We have instead investigated what would happen with the only loop extrusion activity, i.e., in presence of chromatin loops without bridges. Below we show the results for condensin knock-out and common fragile sites from simulations where bridges have been removed, so that there are only looping condensins. In other words, in these new simulations mitotic chromosomes are depicted as a sequence of consecutive loops, in line with our original model, but bridges (green beads in Fig.1 of the manuscript) are no longer inserted.

1. **Condensin knock-out.** In Fig. 4 of the manuscript, as in the original version, we model condensin I and II knock-out. In both cases we vary both activity types: we reduce bridging activity and increase looping activity for condensin I knock-out, while we increase bridging activity and decrease looping activity for condensin II knock-out. This is a hypothesis, as has been clarified in the text (see above), and is based on the idea mentioned at different places in the

manuscript that condensin I may have more bridging activity with respect to condensin II.

Following the Reviewer's question, we have now investigated with new simulations what happens with only the looping activity. In these new simulations we model condensin I and II knock-out simply by inserting longer or shorter chromatin loops respectively, but without including condensin bridges. The loop sizes used in the control simulations, in the condensin I knock-out simulations and in the condensin II knock-out simulations are the same as the ones used in the corresponding three cases examined in Fig.4 where bridges are inserted. As in Fig.4, we quantify the structural changes by computing the width of the chromosomes. Results are reported in Fig. Rep4. On the left (panel (A)) results from the original manuscript (i.e. with bridges) are shown, while on the right (panel (B)) the results from the new simulations (without bridges) are reported. We can see that the absence of bridges results in a larger width in all three cases (control, condensin I knock-out and condensin II knock-out). However, it is also evident how the lack of bridges results in less differences for the control and the condensin II knock-out (blue and green bars in panel (B)). In conclusion, bridges seem to play an important role to enhance the difference in width in the three scenarios, which is also observed experimentally.

Figure Rep4. (A) Average chromosome width from simulations including condensin bridges and for the control, condensin I knock-out and condensin II knock-out set-up. Condensin I knock-out is simulated by introducing longer chromatin loops and reducing the number of condensin bridges, while in the condensin II knock-out shorter loops and higher bridging activity are used. (B) Same results and set-ups as in panel (A), but without inserting bridging activity.

2. **Common fragile sites.** We also performed additional simulations to investigate chromosome structure at common fragile sites in absence of condensin bridges. The new simulations use exactly the same parameters as the simulations in Fig.5 (now Fig. 6) of the main text, with the only difference that

bridges are now absent. First, we discuss the *irregular FISH phenotype* case (Fig. Rep5(A)). To quantify the difference between the original simulations (with bridges) and the new simulations (without bridges), we compute the average 3D distance between the two FISH probes (violet segments in Fig. Rep5 (A,1)) in the two cases. In Fig. Rep5(A,2), we see that the absence of bridges results in a 3D distance only slightly smaller than the one obtained by inserting bridges, meaning that a defect in the looping activity is enough to recapitulate the experimental FISH observations. On the other hand, the characteristic shape taken by chromosomes affected by cytological lesions seems to depend on the presence of condensin bridges. In Fig. Rep5(B1) we report the result of simulations where a cytological lesion has been modelled through a local lack of chromatin loops. In Fig. Rep5(B,1) we see a snapshot from the original simulations including condensin bridges (as in the old Fig.5 of the original manuscript, now Fig. 6), while in Fig. Rep5(B,2) we see a snapshot from a new simulation where two chromatin loops have been removed (as in Fig. Rep5(B,1)), but bridges are not inserted. We notice that the absence of bridges does not lead to the formation of the typical break observed in experiments, suggesting that a defect in the looping activity without bridging is not enough to explain the appearance of cytological lesions.

We thank the Reviewer for prompting us to perform these additional simulations which we believe have strengthened our message and shown more clearly the relevance and importance of a combined bridging and looping activity of condensin. We have now included Figures describing the new simulations shown in Figs. Rep4, Rep5 (as Figs. 5 and 7 of the amended version), and discussed the results in the main text.

Figure Rep5. (A-1) Snapshot showing the two probes (violet segments) whose 3D distance is monitored in our simulations reproducing FISH experiments for irregular phenotypes. **(A-2)** Average 3D distance between the two probes of panel (1) in simulations including and excluding bridging activity (blue and red bar respectively). The case without bridges shows a slightly smaller distance, but no evident changes in the phenotypes are visible. **(B)** Investigation of cytological lesions through simulations where some chromatin loops have been removed and bridging activity can be either present (panel (1)) or absent (panel (2)). The lack of bridges results in a smaller compaction of the mitotic chromosome and the cytological lesion does not become evident as in the case including bridges.

Other comments

Figure 1 is confusing. If blue is chromatin fiber, why is it not looped as written on line 3, first paragraph, results section? What are the grey arcs then? What does it mean "grey arcs, modelled as springs". What do springs refer to?

We take this point, and have redrawn the sketch used in Fig.1. In the new version grey arcs (which used to represent looping interactions among loop roots) are not used anymore and the structure of the bottlebrush polymer is more visible and straightforward.

Figure 1. How was the experimental image of the condensin obtained? If it is from a different work, from which one?

We have obtained this image in the lab. It shows a typical microscopy image of the condensin backbone in mitotic chromosomes, and we have included it to qualitatively support our simulation results. We have now clarified this and included a brief methodology section to describe how the image was taken.

It is difficult to read the panels with four gradations of levels that are referred to as Figure 2, panel C, subpanel (ii), image (top), (left), (bottom), (center), etc. It would be very helpful if each panel were referred to by its own index. 2a, 2b, etc.

Agreed. In the revised version of Fig.2, the panel labelling has changed and each sub-panel has now its own label. We hope this will simplify the understanding of the manuscript.

Figure 2B. What are the different color curves? What model parameters correspond to the each color? Authors should include the table of model parameters. For each parameter, the range of values should be indicated, and it should be clear when parameters are estimated from experiments and when they are variable.

We have added the parameters referring to the several curves shown in Fig.2. Parameters have also been added in Fig.3. In the manuscript we have also specified what parameters come from experiments (such as the loop size) and which ones instead are specific of our simulation set-up (such as the potential representing topoisomerase activity).

Figure 2C(ii). The comparison between the experimental and theoretical curves should be presented on the same graph with the same axes. As it is shown here, it is almost impossible to compare the curves.

We have now modified the last two panels in Fig. 2 showing the comparison between the experimental and simulated probability of contact in mitotic chromosomes. The experimental curves are now adapted from Naumova et al, Science, 2013. Moreover, the axes of the two panels are now represented with the same units (bp).

For the publication in the Journal of Cell Biology, physical quantities in figures should be given in physical units. Time should be in seconds (or minutes, etc, but not relative simulation time), extension should be given in nanometers, microns, etc. Cases where it is not possible should be explained. Model parameters used to simulate curves should be clearly indicated on the figure.

Plots showing contact probabilities in Fig. 2C have now changed and the x-axis units are the same both for experiments and simulations such as it is easier comparing the two results.

Also, all figures have now been remade so as to show quantities in physical units. This was done by using the mapping from simulation to physical units explained in the Materials and Methods section. Parameters are specified in the latter section as well as in figures and captions.

July 18, 2023

RE: JCB Manuscript #202209113R

Prof. Davide Marenduzzo
University of Edinburgh
Peter Guthrie Tait Road
Edinburgh EH9 3FD
United Kingdom

Dear Prof. Marenduzzo:

Thank you for submitting your revised manuscript entitled "Bridging-mediated compaction of mitotic chromosomes." The manuscript was assessed by two of the original reviewers and we would be happy to publish your paper in JCB pending final revisions necessary to address the remaining minor comments and to meet our formatting guidelines (see details below).

A. MANUSCRIPT ORGANIZATION AND FORMATTING:

1) Text limits: Character count for Articles is < 40,000, not including spaces. Count includes title page, abstract, introduction, results, discussion, and acknowledgments. Count does not include materials and methods, figure legends, references, tables, or supplemental legends.

2) Figure formatting: Articles may have up to 10 main text figures. Scale bars must be present on all microscopy images, including inset magnifications. Please avoid pairing red and green for images and graphs to ensure legibility for color-blind readers. If red and green are paired for images, please ensure that the particular red and green hues used in micrographs are distinctive with any of the colorblind types. If not, please modify colors accordingly or provide separate images of the individual channels.

3) Statistical analysis: Error bars on graphic representations of numerical data must be clearly described in the figure legend. The number of independent data points (n) represented in a graph must be indicated in the legend. Please, indicate whether 'n' refers to technical or biological replicates (i.e. number of analyzed cells, samples or animals, number of independent experiments). If independent experiments with multiple biological replicates have been performed, we recommend using distribution-reproducibility SuperPlots (please see Lord et al., JCB 2020) to better display the distribution of the entire dataset, and report statistics (such as means, error bars, and P values) that address the reproducibility of the findings.

Statistical methods should be explained in full in the materials and methods. For figures presenting pooled data the statistical measure should be defined in the figure legends. Please also be sure to indicate the statistical tests used in each of your experiments (both in the figure legend itself and in a separate methods section) as well as the parameters of the test (for example, if you ran a t-test, please indicate if it was one- or two-sided, etc.). Also, if you used parametric tests, please indicate if the data distribution was tested for normality (and if so, how). If not, you must state something to the effect that "Data distribution was assumed to be normal but this was not formally tested."

4) Title: To convey the advance more clearly and increase the accessibility for a broad audience and non-experts we suggest the following title: "Bridging condensins mediate compaction of mitotic chromosomes."

5) Materials and methods: Should be comprehensive and not simply reference a previous publication for details on how an experiment was performed. Please provide full descriptions (at least in brief) in the text for readers who may not have access to referenced manuscripts. The text should not refer to methods "...as previously described." Please provide full details of the cell preparations, analysis of common fragile sites, and the FISH experiments.

6) For all cell lines, vectors, constructs/cDNAs, etc. - all genetic material: please include database / vendor ID (e.g., Addgene, ATCC, etc.) or if unavailable, please briefly describe their basic genetic features, even if described in other published work or gifted to you by other investigators (and provide references where appropriate). You must also indicate in the methods the source, species, and catalog numbers/vendor identifiers (where appropriate) for all of your antibodies, including secondary. If antibodies are not commercial, please add a reference citation if possible.

- 7) Microscope image acquisition: The following information must be provided about the acquisition and processing of images:
- Make and model of microscope
 - Type, magnification, and numerical aperture of the objective lenses
 - Temperature
 - Imaging medium
 - Fluorochromes
 - Camera make and model
 - Acquisition software
 - Any software used for image processing subsequent to data acquisition. Please include details and types of operations involved (e.g., type of deconvolution, 3D reconstitutions, surface or volume rendering, gamma adjustments, etc.).
- 8) References: There is no limit to the number of references cited in a manuscript. References should be cited parenthetically in the text by author and year of publication. Abbreviate the names of journals according to PubMed.
- 9) Supplemental materials: There are strict limits on the allowable amount of supplemental data. Articles may have up to 5 supplemental figures and 10 videos. Please also note that tables, like figures, should be provided as individual, editable files. A summary of all supplemental material should appear at the end of the Materials and methods section. Please include one brief sentence per item.
- 10) Videos: JCB requires require MP4 files for publication. For optimal compatibility across operating systems and devices, please select H.264 compression when saving. Videos may be no larger than 20 MB. The video legends should describe what is being shown, the imaging method, what each color represents, how often frames were collected, the frames/second display rate, and the number of any figure that has related video stills or images.
- 11) eTOC summary: A ~40-50 word summary that describes the context and significance of the findings for a general readership should be included on the title page. The statement should be written in the present tense and refer to the work in the third person. It should begin with "First author name(s) et al..." to match our preferred style.
- 12) Conflict of interest statement: JCB requires inclusion of a statement in the acknowledgements regarding competing financial interests. If no competing financial interests exist, please include the following statement: "The authors declare no competing financial interests." If competing interests are declared, please follow your statement of these competing interests with the following statement: "The authors declare no further competing financial interests."
- 13) A separate author contribution section is required following the Acknowledgments in all research manuscripts. All authors should be mentioned and designated by their first and middle initials and full surnames. We encourage use of the CRediT nomenclature (<https://casrai.org/credit/>).
- 14) ORCID IDs: ORCID IDs are unique identifiers allowing researchers to create a record of their various scholarly contributions in a single place. At resubmission of your final files, please consider providing an ORCID ID for as many contributing authors as possible.
- 15) Journal of Cell Biology now requires a data availability statement for all research article submissions. These statements will be published in the article directly above the Acknowledgments. The statement should address all data underlying the research presented in the manuscript. Please visit the JCB instructions for authors for guidelines and examples of statements at (<https://rupress.org/jcb/pages/editorial-policies#data-availability-statement>).

B. FINAL FILES:

****The license to publish form must be signed before your manuscript can be sent to production. A link to the electronic license to publish form will be sent to the corresponding author only. Please take a moment to check your funder requirements before choosing the appropriate license.****

Thank you for this interesting contribution, we look forward to publishing your paper in Journal of Cell Biology.

Sincerely,

Arshad Desai, PhD
Monitoring Editor
Journal of Cell Biology

Dan Simon, PhD
Scientific Editor
Journal of Cell Biology

Reviewer #2 (Comments to the Authors (Required)):

Authors have addressed all my comments. I recommend publication.

Reviewer #3 (Comments to the Authors (Required)):

In the revised version of the manuscript Forte et al., have addressed all my major concerns. The new manuscript explains assumptions of the proposed model and discusses its limitations. There are two small things I would suggest addressing before publishing:

1. By introducing "multivalent" interactions authors assume that bridging condensin can bind more than one chromatin bead. It would be helpful to quantify how many DNAs are bound to bridging condensins in steady state in a typical simulation. If it's random, than seeing a distribution for simulations in figures 2B,C would help to understand what actually happens mechanistically at the molecular level.
2. I could not understand why in a simulation without bridging, the polymer after extension does not retract back to its original length. If loops are fixed and stay where they were and don't change in length, what changes in the polymer after the force is applied and then removed? It would be good to clarify this.

Otherwise, I think it is ready for publication.

Dear Editor,

Many thanks for sending us your decision and the Reviewers' comments on our manuscript, "Bridging condensins mediate compaction of mitotic chromosomes" (note changed title, and thanks for your suggestion in this regard).

We are glad that all Reviewers now support publication. We have addressed the final minor suggestions by Reviewer 3 in the current, slightly revised version (changes are highlighted in blue for convenience). We also provide a point-by-point reply to Reviewers below. Note that in this version we have also incorporated all Editorial suggestions and requests, in particular we have edited the Methods Section to conform to the journal formatting guidelines, expanding the discussion of the FISH methods, instead of referring to previous papers.

We hope that the current version can now be published in J. Cell. Biol.

With best wishes,

G. Forte, L. Boteva, F. Conforto, N. Gilbert, P. R. Cook, D. Marenduzzo

Reply to Reviewers

Reply to Reviewer 2's second report

Authors have addressed all my comments. I recommend publication.

We thank the Reviewer for their careful reading, and are glad that they support publication of our manuscript.

Reply to Reviewer 3's second report

In the revised version of the manuscript Forte et al., have addressed all my major concerns. The new manuscript explains assumptions of the proposed model and discusses its limitations. There are two small things I would suggest addressing before publishing:

[...]

Otherwise, I think it is ready for publication.

We thank the Reviewer for their careful reading of our manuscript. We are glad that they found our revision addressed all their major concerns and that they find the manuscript ready for publication subject to two minor points, which we have addressed in the current version, as discussed below in our point-by-point reply.

1. By introducing "multivalent" interactions authors assume that bridging condensin can bind more than one chromatin bead. It would be helpful to quantify how many DNAs are bound to bridging condensins in steady state in a typical simulation. If it's random, than seeing a distribution for simulations in figures 2B,C would help to understand what actually happens mechanistically at the molecular level.

We agree it is useful to quantify the degree of multivalence in practice, and have now provided the average number of chromatin beads bound by a condensin bridge (in the first paragraph of the section “Condensin-mediated bridging compacts bottlebrushes into cylinders”). We have measured this number by analysing steady-state configurations from our simulations (as there is no prescribed valence in our model).

2. I could not understand why in a simulation without bridging, the polymer after extension does not retract back to its original length. If loops are fixed and stay where they were and don't change in length, what changes in the polymer after the force is applied and then removed? It would be good to clarify this.

This is a good point, and we also wondered about this. Our interpretation is that the difference between the length of the cylinders before and after pulling is likely to be due to the initial condition from which we start. In particular, the polymer may be stuck in a slightly crumpled, very long-lived, metastable phase before the pulling simulation. We believe this situation may actually be realistic, as polymers inside the nucleus are also not in equilibrium (see, e.g., Rosa and Everaers, PLoS Comp. Biol. e1000153 (2008)). We have now mentioned the potential role of the initial condition for this result when discussing the stretching simulations.